# Aven recognition of RNA G-quadruplexes regulates translation of the *mixed lineage leukemia* protooncogenes

**Palaniraja Thandapani[1,2,6,7†], Jingwen Song[1,2,6,7†], Valentina Gandin[1], Yutian Cai[1,3], Samuel G Rouleau[4], Jean-Michel Garant[4], Francois-Michel Boisvert[5], Zhenbao Yu[1,2,6,7], Jean-Pierre Perreault[4], Ivan Topisirovic[1,3,6], Stéphane Richard[1,2,6,7]\***

[1]Terry Fox Molecular Oncology Group, Segal Cancer Center, Jewish General Hospital, Montréal, Canada; [2]Bloomfield Center for Research on Aging, Lady Davis Institute for Medical Research, McGill University, Montréal, Canada; [3]Department of Biochemistry, McGill University, Montréal, Canada; [4]Département de Biochimie, Université de Sherbrooke, Sherbrooke, Canada; [5]Département d'Anatomie et de Biologie Cellulaire, Faculté de Médecine et des Sciences de la Santé, Pavillon de Recherche Appliquée au Cancer, Université de Sherbrooke, Sherbrooke, Canada; [6]Department of Oncology, McGill University, Montréal, Canada; [7]Department of Medicine, McGill University, Montréal, Canada

**Abstract** G-quadruplexes (G4) are extremely stable secondary structures forming stacks of guanine tetrads. DNA G4 structures have been extensively studied, however, less is known about G4 motifs in mRNAs, especially in their coding sequences. Herein, we show that Aven stimulates the mRNA translation of the mixed lineage leukemia (*MLL*) proto-oncogene in an arginine methylation-dependent manner. The Aven RGG/RG motif bound G4 structures within the coding regions of the MLL1 and MLL4 mRNAs increasing their polysomal association and translation, resulting in the induction of transcription of leukemic genes. The DHX36 RNA helicase associated with the Aven complex and was required for optimal translation of G4 mRNAs. Depletion of Aven led to a decrease in synthesis of MLL1 and MLL4 proteins resulting in reduced proliferation of leukemic cells. These findings identify an Aven-centered complex that stimulates the translation of G4 harboring mRNAs, thereby promoting survival of leukemic cells.

**\*For correspondence:** stephane.richard@mcgill.ca

[†]These authors contributed equally to this work

**Competing interests:** The authors declare that no competing interests exist.

## Introduction

RNA-binding proteins (RBPs) coordinate many steps of RNA metabolism ranging from splicing, RNA processing, RNA transport, mRNA translation, and RNA degradation (*Glisovic et al., 2008*). RBPs associate with specific RNA motifs and/or secondary structures within coding, untranslated regions, and non-coding RNAs in functional units called ribonucleoprotein (RNP) complexes (*Mitchell and Parker, 2014*). Defects in RBPs have been associated with many complex diseases ranging from neurological disorders to cancer (*Lukong et al., 2008*; *Cooper et al., 2009*; *Castello et al., 2013*; *Ramaswami et al., 2013*).

RBPs are predominantly defined by the presence of RNA-binding domains within their sequences (*Chen and Varani, 2013*). Recently, several 'interactome capture' strategies have been performed to identify RBPs genome-wide. In addition to identifying the known RBPs, these approaches have identified numerous mammalian proteins that do not possess a canonical RNA-binding domain

**eLife digest** To make a protein, the DNA sequence that encodes it is first copied to make a molecule of messenger RNA (or mRNA for short). The mRNA is then used as a set of instructions to assemble a protein in a process called translation. Both DNA and RNA molecules can fold into particular shapes. One such structure is known as a G-quartet and involves the DNA or RNA folding back on itself to form a highly stable planar structure. Stacks of G-quartets can form structures known as G-quadruplexes, but little is known about the G-quadruplexes that form in mRNA molecules.

Leukemia affects cells in the bone marrow and causes blood cells to develop abnormally. A protein called Aven is often found in increased amounts in certain types of leukemic cells, but it was not clear how Aven affects how leukemia develops. Thandapani, Song et al. have now found that in leukemic cells, Aven binds to G-quadruplexes found in two mRNA molecules that encode proteins that are linked to leukemia. This binding increases the translation of these mRNAs, with translation occurring most efficiently when a particular enzyme called a helicase—which remodels RNA—also bound to Aven. Reducing the amount of Aven in cells caused fewer of the leukemic proteins to be produced, which also reduced the growth and multiplcation of leukemic cells.

These findings raise the possibility that drugs that disrupt how Aven works could form part of treatments for leukemia. The next challenge will be to identify the signaling pathways that communicate with Aven and to define all the G-quadruplex mRNAs that are regulated by Aven.

(*Baltz et al., 2012*; *Castello et al., 2012*; *Kwon et al., 2013*). Interestingly, RBPs that harbor repeated sequences including YGG and RGG motifs were identified (*Castello et al., 2012*). The RGG/RG motif is enriched in proteins associated with RNA and is a known RNA-binding interface (reviewed in *Thandapani et al., 2013*). The RGG/RG motif, also called RGG box, was shown to bind RNA (*Kiledjian and Dreyfuss, 1992*). Subsequently, the RGG/RG motifs of Nucleolin, FMRP, FUS, and EWS were also shown to bind guanine-rich sequences that are potential G-quadruplexes (*Darnell et al., 2001*; *Takahama et al., 2011*, *2013*; *Haeusler et al., 2014*). RGG/RG motifs also mediate protein–protein interactions. Notably, the RGG/RG motif of yeast Scd6 mediates interactions with eIF-4G, which leads to stress granule formation and inhibition of cap-dependent translation (*Rajyaguru et al., 2012*). Despite these recent advances, little is known about the role of RGG/RG motifs that bind both RNA and proteins.

G-quadruplexes (G4) are planar structures of stacks of guanine tetrads stabilized by monovalent potassium or sodium ions. G-quadruplexes have been shown to regulate DNA replication, DNA repair, gene expression, and telomeres (*Bates et al., 2007*). Less is known about G4 structures found in RNA. There are >1500 potential G4s (PG4s) in 5′-UTR of mRNAs alone (*Beaudoin and Perreault, 2010*), but not all PG4s form stable G-quadruplexes, which are influenced by the numbers of G-quartets, the possibility of bulge formation, the length of the loops, and the presence of alternative Watson–Crick base pair-based stable structure (*Burge et al., 2006*; *Mukundan and Phan, 2013*; *Jodoin et al., 2014*). Some PG4s in the 5′-UTR of mRNAs form *bona fide* G-quadruplexes and inhibit cap-dependent translation (*Kumari et al., 2007*; *Beaudoin and Perreault, 2010*; *Bugaut and Balasubramanian, 2012*). Recently, inhibitors of DEAD box RNA helicase eIF4A or eIF4A1 depletion have been shown to selectively inhibit translation of mRNAs with G-quadruplexes in their 5′ UTR (*Wolfe et al., 2014*; *Modelska et al., 2015*). However, presence of G-quadruplexes in 5′ UTRs does not appear to be sufficient to render translation of mRNAs sensitive to changes in eIF4A activity (*Rubio et al., 2014*). In addition to the incomplete understanding of the role of 5′ UTR G-quadruplexes in translation control, little is known about how G4 structures in open-reading frames (ORFs) affect translation.

Arginine residues within RGG/RG motifs are preferred substrates for methylation by protein arginine methyltransferases (PRMTs) (*Thandapani et al., 2013*). Arginine methylation is known to regulate many cellular processes including signal transduction, transcription, pre-mRNA splicing, and DNA repair (*Bedford and Richard, 2005*; *Bedford and Clarke, 2009*; *Xu et al., 2013*). PRMT1 generates >85% of asymmetric dimethylarginines found in cells with preference for RGG/RG motif containing proteins (*Bedford and Clarke, 2009*). PRMT1 is known for its nuclear roles in regulating gene expression and DNA damage (*Strahl et al., 2001*; *Wang et al., 2001*; *An et al., 2004*;

*Boisvert et al., 2005*), however, less is known about its cytoplasmic roles. PRMT1-deficient mice die at E6.5 and the absolute removal of PRMT1 in mouse embryo fibroblasts (MEFs) leads to cell death (*Pawlak et al., 2000*; *Yu et al., 2009*). To identify other biological processes regulated by arginine methylation, we performed a bioinformatics approach to identify proteins harboring RGG/RG motifs and one such protein we identified was Aven (*Thandapani et al., 2013*).

Aven is a predominantly cytoplasmic protein required for cell survival and it has been shown to function as an apoptotic inhibitor by interaction with and stabilizing the pro-survival protein Bcl-x$_L$, as well as inhibiting the function of Apaf-1 (*Chau et al., 2000*). It was proposed that the proteolytic cleavage of Aven by Cathepsin D is required for its anti-apoptotic activity (*Melzer et al., 2012*). Furthermore, Aven is required for ataxia telangiectasia-mutated (ATM) activation in *Xenopus* oocytes and HeLa cells (*Guo et al., 2008*) and ataxia telangiectasia-related activation following DNA damage in osteosarcoma cells (*Baranski et al., 2015*). High Aven expression correlates with poor survival in metastatic patients with osteosarcomas (*Baranski et al., 2015*). The elevated Aven expression is also frequently observed in acute myeloid leukemia and acute lymphoblastic leukemia (T-ALL) and is associated with poor prognosis (*Paydas et al., 2003*; *Choi et al., 2006*). A transgenic mouse model with T cell-specific overexpression of Aven showed that its expression enhanced T-cell lymphoma-genesis in the absence of p53 (*Eismann et al., 2013*). The mechanism by which Aven promotes hematological malignancies is yet to be understood.

Herein, we report that the methylation of the RGG/RG motif of Aven functions in the translational control of mRNAs harboring G4 structures in their ORFs. The association of Aven with polysomes was dependent on the arginine methylation of its RGG/RG motif and on the methyl-dependent interactions with the Tudor domains of SMN and TDRD3, previously shown to be associated with polysomes (*Goulet et al., 2008*; *Sanchez et al., 2013*). We identify Aven to be an RBP, as its RGG/RG motif bound G4 motifs in the ORFs of mRNAs encoding the mixed lineage leukemia (MLL) family proteins MLL1 and MLL4. The RGG/RG motif of Aven also associated with the G4 RNA helicase, DHX36, and this helicase was required for optimal translation of Aven-regulated mRNAs. Furthermore, Aven-deficient T-ALL cell lines had reduced MLL1 and MLL4 protein levels, but not mRNA levels, which were paralleled by proliferation defects. These findings define a hitherto unknown mechanism of action for arginine methylation in regulating translation of a subset of mRNAs including those encoding pivotal leukemogenic transcriptional regulators MLL1 and MLL4.

## Results

### Aven RGG/RG motif is methylated by PRMT1

Aven harbors an N-terminal RGG/RG motif (*Thandapani et al., 2013*), a nuclear export sequence (*Esmaili et al., 2010*), and a predicted BH3 domain (*Hawley et al., 2012*) (*Figure 1A*). To define the function of the Aven RGG/RG motif, we initially investigated whether the motif was methylated by protein arginine methyltransferase 1 (PRMT1). An in vitro methylation assay was performed using a glutathione-S-transferase (GST)-Aven RGG/RG fusion protein incubated with recombinant GST-PRMT1 in the presence of (*methyl-*$^3$H)-S-adenosyl-L-methionine. The proteins were separated by SDS-PAGE, stained with Coomassie Blue to visualize loading, and the methylated proteins were observed by fluorography. The GST-AvenRGG/RG fusion protein migrated as a doublet and was methylated by PRMT1, while GST alone was not methylated (*Figure 1B*, lanes 5, 6).

We next examined whether Aven and PRMT1 associated in vivo. HEK293T cells were transfected with Myc-epitope tagged Aven or empty vector (pcDNA3.1) and cell extracts were immunoprecipitated (IP) with anti-Myc antibodies. The IP proteins were separated by SDS-PAGE and immunoblotted with anti-PRMT1 antibodies (*Figure 1C*). PRMT1 was present in anti-Myc immunoprecipitates of the Myc-Aven-transfected cells, but not in the empty vector-transfected cells (*Figure 1C*, lanes 3, 4). Immunoblotting with anti-Myc confirmed the presence of Myc-Aven (*Figure 1C*, lanes 7, 8). To examine whether Aven is a substrate of PRMT1 in vivo, we depleted HEK293T cells of PRMT1 using siRNA. The cells were also transfected with empty plasmid (pcDNA3.1) or an expression vector encoding 5 tags of the Myc epitope linked to Aven (Myc-Aven). Cellular lysates were IP with immunoglobulin (IgG) control or anti-Myc antibodies, resolved by SDS-PAGE, and immunoblotted with ASYM25b, an asymmetric dimethylarginine-specific antibody. We observed that Myc-Aven was arginine methylated in *PRMT1*-proficient, but not in PRMT1-deficient cells (*Figure 1D*, compare lanes 3 and 4). An anti-Myc immunoblot confirmed the immunoprecipitations (*Figure 1D*, lanes 7 and 8). Immunoblots of total cell

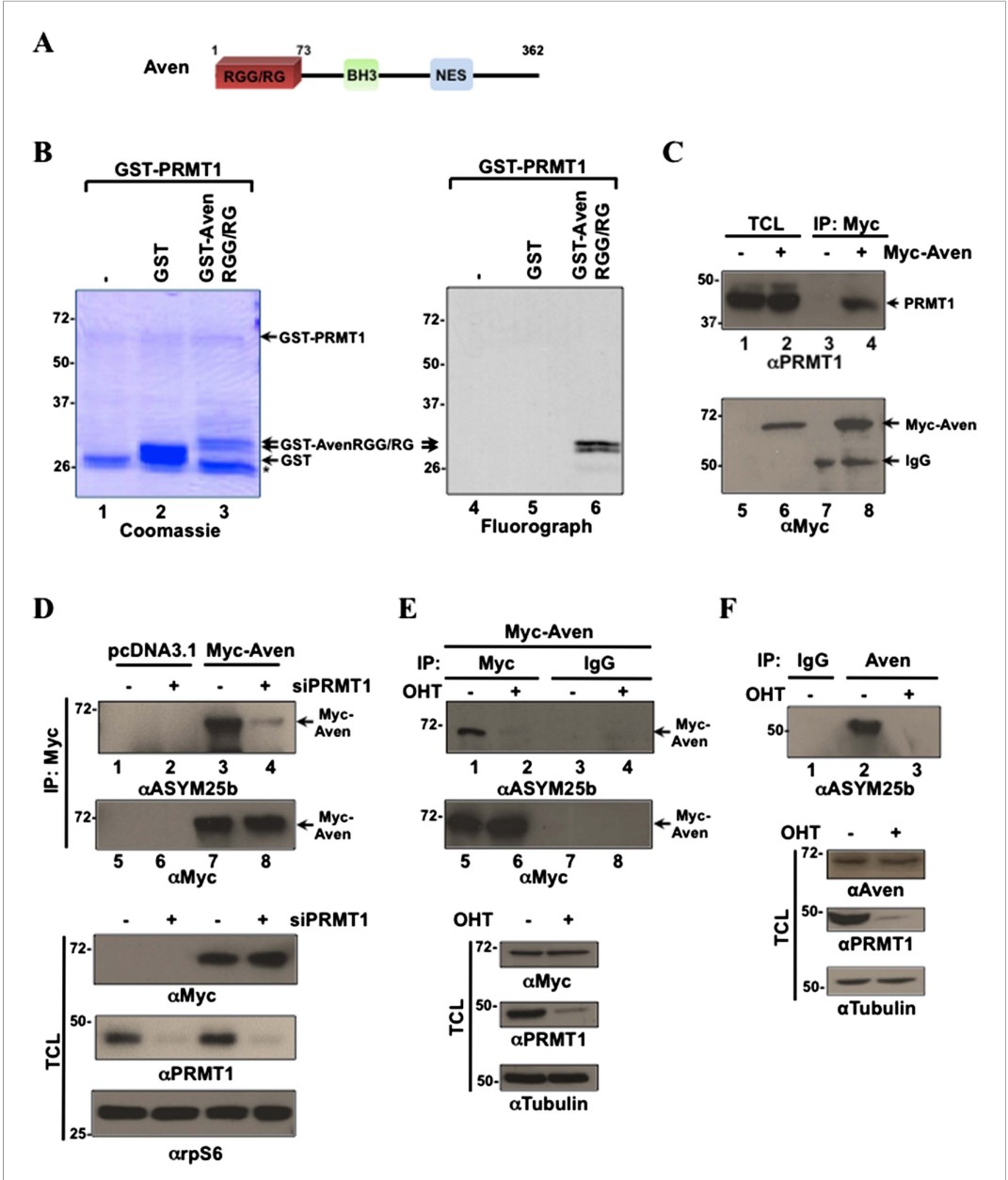

**Figure 1.** Aven is a substrate of PRMT1. (**A**) Schematic diagram of Aven with its RGG/RG motif, putative BH3 domain, and nuclear export sequence (NES). (**B**) In vitro methylation assay with GST-PRMT1 and GST-AvenRGG/RG with ($^3$H)-S-adenosyl-L-methionine as the methyl donor (n > 4). Proteins were resolved by SDS-PAGE, stained with Coomassie Blue (left), and analyzed by fluorography (right). The migration of the molecular mass markers is shown on the left in kDa and the migration of the GST-PRMT1, glutathione-S-transferase (GST), GST-AvenRGG/RG proteins is indicated with arrows. The asterisk (*) denotes degraded proteins from the GST-PRMT1 preparation.

(**C**) HEK293T cells transfected with empty vector pcDNA3.1 (lanes 1, 3, 5, 7) or Myc-Aven-pcDNA3.1 (lanes 2, 4, 6, 8) were lysed and immunoprecipitated (IP) with anti-Myc antibodies (lanes 3, 4, and 7, 8). The total cell lysates (TCL) and the bound proteins were resolved by SDS-PAGE and immunoblotted with anti-PRMT1 (lanes 1–4) or −Myc antibodies (5–8). TCL denotes input TCL and IgG represents the heavy chain of immunoglobulin G.

(**D**) HEK293T cells were cotransfected with siGFP (-siPRMT1) or siPRMT1 (+siPRMT1) and pcDNA3.1 or Myc-Aven plasmids. After 48 hr, the cells were lysed and IP with anti-Myc antibodies, the proteins were resolved by SDS-PAGE and immunoblotted with anti-ASYM25b (lanes 1–4), and anti-Myc antibodies (lanes 5–8). TCL were immunoblotted with anti-Myc, anti-PRMT1, and anti-rpS6 antibodies. The latter was to control for equal loading. (**E**) PRMT1$^{FL/−;CreERT}$ mouse embryo fibroblasts (MEFs) treated with 4-hydroxytamoxifen (OHT) for 6 days or left untreated were

*Figure 1. continued on next page*

*Figure 1. Continued*

transfected with Myc-Aven followed by anti-Myc antibody immunoprecipitations and the methylation monitored by immunoblotting with ASYM25b (lanes 1–4) or anti-Myc antibodies (lanes 5–8). TCL were immunoblotted with anti-PRMT1, anti-Aven, and anti-tubulin antibodies as indicated. (**F**) $PRMT1^{FL/-;CreERT}$ MEFs treated with OHT for 6 days or left untreated were lysed and IP with anti-Aven (ab77014) or IgG antibodies. Immunoprecipitates were blotted with ASYM25b. TCL were immunoblotted with anti-PRMT1, anti-Aven, and anti-tubulin antibodies as indicated.

The following figure supplement is available for figure 1:

**Figure supplement 1**. Aven harbors dimethylarginines within its RGG/RG motif.

---

lysates confirmed the myc-Aven expression and the PRMT1 knockdown and rpS6 was used as a loading control (*Figure 1D*, lower panels).

A similar experiment was performed using conditional $PRMT1^{FL/-;CreERT}$ MEFs (*Yu et al., 2009*). Ablation of PRMT1 was achieved by treating the cells with 4-hydroxytamoxifen (OHT) for 6 days. An expression vector encoding Myc-Aven was transfected into control ($PRMT1^{FL/-;CreERT}$; –OHT) and PRMT1-deficient ($PRMT1^{FL/-;CreERT}$; +OHT) MEFs. Myc-Aven was arginine methylated in *PRMT1*-proficient, but not in PRMT1-deficient cells (*Figure 1E*, compare lanes 1 and 2). We next proceeded to show that endogenous Aven is arginine methylated. Conditional $PRMT1^{FL/-;CreERT}$ MEFs were used for Aven immunoprecipitations and indeed endogenous Aven was asymmetrically dimethylated by PRMT1 (*Figure 1F*). Endogenous Aven migrates at ~50 kDa (*Figure 1F*), while myc-Aven was generated with multiple Myc tags to avoid overlap with heavy chain of IgG during immunoprecipitation and migrated at ~70 kDa (*Figure 1C–E*). To identify the arginines within the RGG/RG motif that are methylated, immunopurified Myc-Aven was subjected to mass spectrometry analysis. We identified the dimethylation of Aven R37, R63, and R66 (*Figure 1—figure supplement 1*). Moreover, we also identified the dimethylation of R8, R11, R50, as well as the monomethylation of R5, R8, R28, and R37. Arginines R5, R8, R11, R37, R50, and R63 are also conserved in murines (*Figure 1—figure supplement 1*). Taken together, these findings demonstrate that the Aven RGG/RG motif is methylated by PRMT1 on conserved arginines.

## RGG/RG motif of Aven binds G4 sequences

To define the role of the Aven RGG/RG motif, we generated a mutant that lacks the motif by deleting amino acids 1 to 73. AvenΔRGG was not recognized by ASYM25b, confirming that all the methylarginines reside in the N-terminus of Aven (*Figure 2A*). AvenΔRGG was able to activate ATM, comparably to wild-type Aven and deletion of the RGG/RG motif did not interfere with its ability to oligomerize (*Figure 2—figure supplement 1*). Aven is predominantly localized in the cytoplasm with some weak nuclear staining (*Figure 2—figure supplement 1*), as reported previously (*Chau et al., 2000*), and FLAG-AvenΔRGG had the same cellular localization pattern as wild-type Aven (*Figure 2—figure supplement 1*).

RGG/RG motifs are enriched amongst RBPs and they possess inherent RNA-binding activity (reviewed in *Thandapani et al., 2013*). A high-affinity RNA sequence that forms a G4, termed *sc1*, binds RGG/RG sequences (*Phan et al., 2011*). To test whether the RGG/RG motif of Aven binds G4 sequences, we performed binding assays using *sc1*. Biotinylated RNA sequences of *sc1* were generated, heated, and slowly cooled in the presence of $K^+$ to favor formation of the G4 RNA structure. The RNA was used in an affinity 'pull-down' assay with HEK293T cell lysates. The presence of Aven after different washes of sodium chloride was monitored by immunoblotting following separation of the bound proteins by SDS-PAGE. Aven bound the *sc1* wild-type G4 RNA, but not a mutant *sc1* RNA sequence that is predicted not to form the G4 structure (*Figure 2B*).

To determine whether the Aven binding to the G4 RNA structures was mediated by the RGG/RG motif, binding assays were performed in HEK293T cells expressing FLAG-Aven and the FLAG-AvenΔRGG. FLAG-Aven, but not FLAG-AvenΔRGG, bound the *sc1* G4 structure indicating that the RGG/RG motif is necessary for binding (*Figure 2C*). We subsequently investigated whether methylation of the RGG/RG motif influences binding to the *sc1* G4 structure. Hypomethylated Aven was obtained by depleting HEK293T cells of PRMT1 with siRNAs. Aven bound equally well to biotinylated *sc1* G4

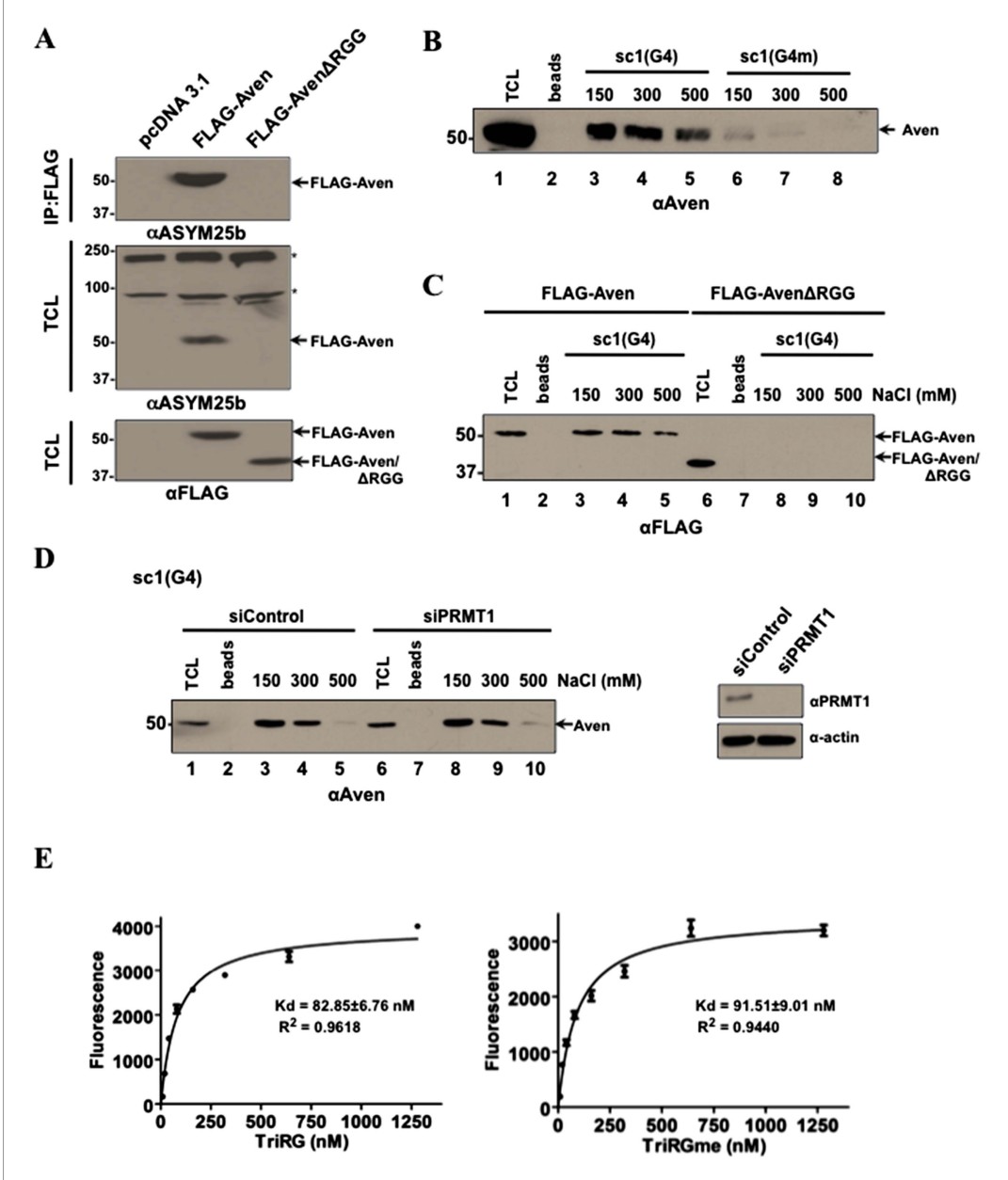

**Figure 2**. Aven binds G4 RNA sequences in an arginine methylation independent manner. (**A**) U2OS cells transfected with pcDNA3.1, or expression vectors encoding FLAG-Aven or FLAG-AvenΔRGG were IP with anti-FLAG antibodies and immunoblotted with ASYM25b or anti-FLAG antibodies as indicated. The molecular mass markers are shown on the left in kDa and the migration of FLAG-Aven and FLAG-AvenΔRGG is shown. The asterisks (*) denote unknown arginine methylated proteins. (**B**) Biotinylated *sc1* G4 bound to Streptavidin were incubated with HEK293T cell lysates. The bound proteins were washed with increasing NaCl (mM) as indicated and visualized by SDS-PAGE followed by immunoblotting with anti-Aven antibodies. (**C**) Biotinylated *sc1* G4 or (G4m) bound to Streptavidin was incubated with cellular lysates expressing FLAG-Aven and FLAG-AvenΔRGG and detected as in panel **B**. (**D**) Biotinylated *sc1* G4 RNA bound to Streptavidin beads was used to pull-down Aven from PRMT1-depleted HEK293T cells. Aven binding was performed as in panel **B**. PRMT1 depletion was confirmed by immunoblotting. (**E**) Biotinylated methylated and unmethylated Aven TriRG peptides were pre-bound on Streptavidin plates and were incubated with fluorescein-labeled *sc1* G4 RNA. The bound RNA was quantified by measuring fluorescence at 521 nm. The experiment was performed twice in triplicates.

The following figure supplement is available for figure 2:

**Figure supplement 1**. Aven RGG/RG motif binds RNA and does not regulate ATM activation, nor Aven cellular localization.

RNA from wild-type or PRMT1-depleted cells (*Figure 2D*). A biotinylated Aven RGG/RG peptide denoted as TriRG was synthesized with or without asymmetric dimethylarginines, and RNA binding was measured using fluorescently labeled RNA. RGG/RG motif whether harboring arginine or asymmetric dimethylarginines bound a fluoresceinated *sc1* G4 RNA, induced to fold into a G4 structure prior to binding, with same relative affinities with a Kd of ~80–90 nM (*Figure 2E*). Another set of peptides spanning the DiRGG motif of Aven with and without asymmetric dimethylarginines also bound the fluoresceinated *sc1* G4 RNA with lower affinity than the TriRG peptides (Kd of ~175–200 nM, *Figure 2—figure supplement 1*), suggesting that several RGG/RG motifs in Aven are able to bind RNA. These finding show that Aven is an RBP that interacts with G4 RNA sequences via its RGG/RG motifs independent of arginine methylation.

## RGG/RG motif of Aven interacts with TDRD3 and SMN in a methyl-dependent manner

Arginine methylation is known to regulate protein–protein interactions with Tudor domain-containing proteins (*Chen et al., 2011*). Although there are many Tudor domain-containing proteins, methylated RGG/RG motifs are known to interact specifically with the Tudor domains of TDRD3, SMN, and SPF30 (*Selenko et al., 2001*; *Cote and Richard, 2005*). Since arginine methylation did not influence the ability of Aven to bind RNA, we next investigated whether Aven interacts with TDRD3, SMN, and SPF30 in a methyl-dependent manner. U2OS cellular lysates expressing FLAG-Aven or FLAG-AvenΔRGG were incubated with the GST-TDRD3, GST-SMN, and GST-SPF30 bound to Sepharose beads. The presence of bound Aven was detected by SDS-PAGE followed with anti-FLAG immunoblotting. FLAG-Aven, but not FLAG-AvenΔRGG, interacted with the GST Tudor domains of TDRD3 and SMN, suggesting that TDRD3 and SMN Tudor domains interact with the Aven RGG/RG motif (*Figure 3A*). GST-SPF30 Tudor domain had a weak interaction with FLAG-Aven, but not with FLAG-AvenΔRGG (*Figure 3A*). To verify whether Aven interacts with TDRD3 and SMN in vivo, co-immunoprecipitations were performed. Endogenous TDRD3 and SMN co-IP with FLAG-Aven, but not with FLAG-AvenΔRGG (*Figure 3B,C*). To verify whether FLAG-Aven interacts with TDRD3 and SMN in an arginine methylation-dependent manner, co-immunoprecipitations were performed in PRMT1-depleted and control HEK293T cells. Indeed, cells depleted of PRMT1 showed reduced interaction between FLAG-Aven and SMN and TDRD3, as compared to the control (*Figure 3D*). To confirm the interaction of endogenous Aven with SMN and TDRD3, conditional *PRMT1$^{FL/-;CreERT}$* MEFs were treated with OHT or a vehicle, and the cellular lysates were IP with anti-Aven antibodies. Immunoprecipitates were resolved by SDS-PAGE and immunoblotted with anti-SMN and anti-TDRD3 antibodies. The ability of SMN and TDRD3 to co-immunoprecipitate with endogenous Aven was lost in PRMT1-deficient cells (+OHT; *Figure 3E*). These findings confirm that methylation of the RGG/RG motif is required for interaction between the Aven/TDRD3 and Aven/SMN complexes.

## Methylated RGG/RG motif regulates association of Aven with polysomes

To identify the interactome of the RGG/RG motif of Aven, we used a stable isotope labeling by amino acids in cell culture (SILAC) approach to quantify protein complexes differentially associated with Aven and AvenΔRGG (*Blagoev et al., 2003*). U2OS cells transfected with pcDNA3.1, FLAG-Aven, or FLAG-AvenΔRGG were light (L), medium (M), or heavy (H) SILAC labeled, respectively, and IP with anti-FLAG antibodies. The data were expressed as fold-enrichment of Aven over control (M/L) and AvenΔRGG over control (H/L). Aven was identified and quantified with high M/L and H/L ratios, while PRMT1 was only found with a high ratio M/L showing as expected that it associates with its RGG-containing substrates (*Figure 4—source data 1*). Of the 146 proteins enriched with Aven, but not AvenΔRGG, ~23% were ribosomal proteins and ~10% were RBPs, their association is likely RNA-dependent (*Figure 4—source data 1*). These data suggested that Aven, but not AvenΔRGG, associates with ribosomal proteins and/or RNP complexes, thereby implying that Aven associates with ribosomes in an RGG/RG motif-dependent manner. To address this question, we first assessed whether endogenous Aven and PRMT1 associate with ribosomes by sedimenting cytoplasmic extracts on 5–50% sucrose gradients by ultracentrifugation (*Figure 4A*). Subsequently, sucrose gradients were fractionated to separate cytoplasmic mRNPs, ribosomal subunits, monosomes, and polysomes (*Gandin et al., 2014*), and the amount of Aven, PRMT1, rpS6 (ribosomal protein S6), and β-tubulin in each fraction was

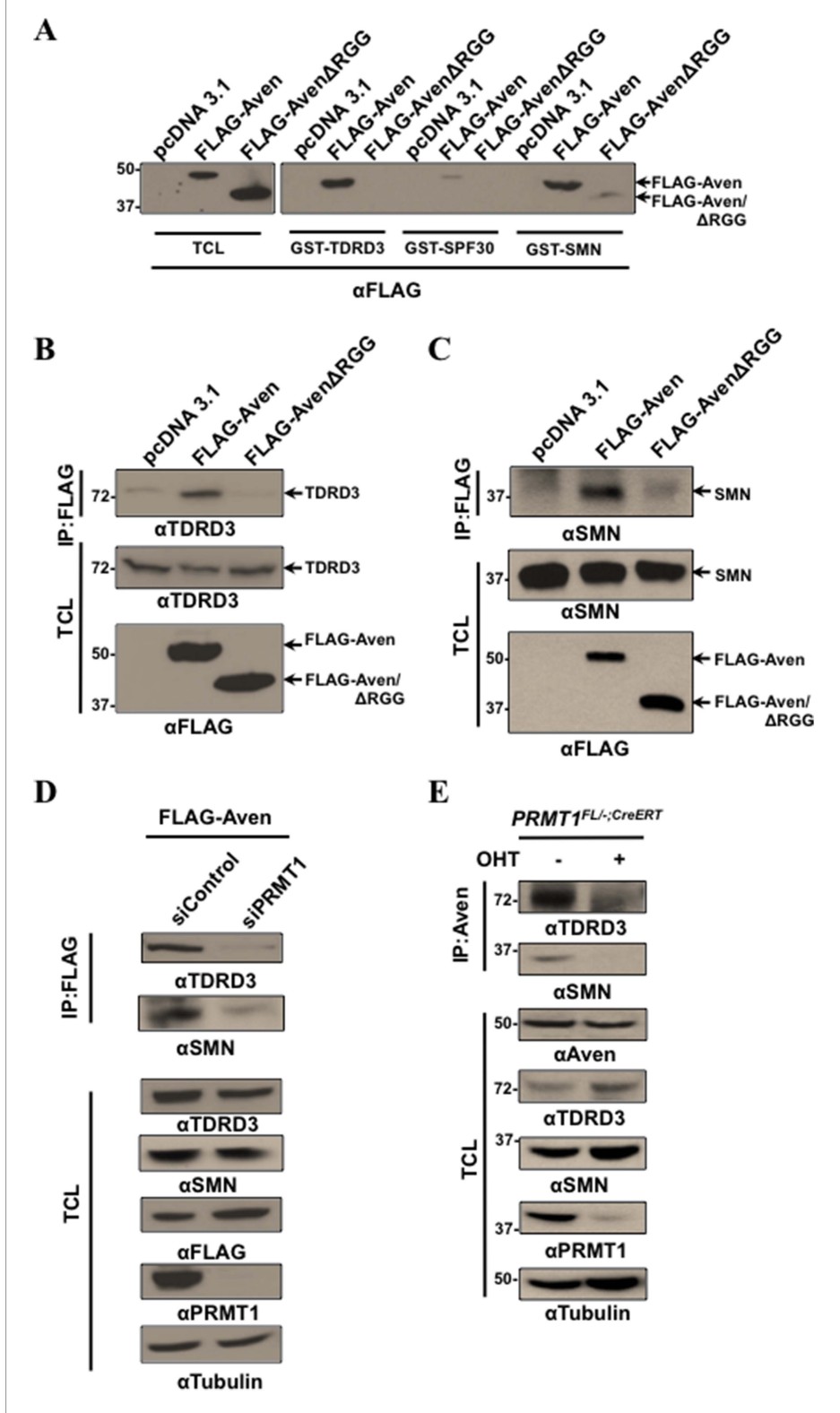

**Figure 3**. Tudor domains of TDRD3 and SMN recognize methylated Aven. (**A**) Recombinant Tudor domains of TDRD3, SPF30, and SMN were fused GST and used in 'pull-down' assays with HEK293T lysates expressing pcDNA3.1 (control), FLAG-Aven, or FLAG-AvenΔRGG. The bound proteins were separated by SDS-PAGE and immunoblotted with anti-FLAG antibodies. (**B**, **C**) Lysates from HEK293T lysates expressing pcDNA3.1 (control),
*Figure 3. continued on next page*

*Figure 3. Continued*

FLAG-Aven, or FLAG-AvenΔRGG were IP with anti-FLAG antibodies. Co-immunoprecipitation of endogenous TDRD3 and SMN was detected by immunoblotting. (**D**) Aven interaction with TDRD3 and SMN was reduced in cells deficient for PRMT1 using siRNAs. FLAG-Aven was co-expressed with either siControl or siPRMT1 in U2OS cells. Anti-FLAG antibody immunoprecipitations were performed and the presence of endogenous TDRD3 and SMN monitored by immunoblotting following separation by SDS-PAGE. (**E**) PRMT1$^{FL/-;CreERT}$ MEFs treated with OHT for 6 days or left untreated were lysed and IP with anti-Aven antibodies. Co-immunoprecipitation of endogenous TDRD3 and SMN was detected by immunoblotting (upper panels).

determined by immunoblotting. Aven and PRMT1 co-sedimented with the heavier polysomal fractions with rpS6, while β-tubulin was restricted to the lighter fractions of the sucrose gradient corresponding to cytoplasmic mRNP fractions (*Figure 4B*, fractions 11 to 16). To investigate whether the RGG/RG motif and its methylation regulates the recruitment of Aven to polysomes, we performed polysomal fractionation in HEK293T cells transfected with FLAG-Aven or FLAG-AvenΔRGG (*Figure 4—figure supplement 1*). FLAG-Aven co-sedimented with the heavier polysomal fractions, while FLAG-AvenΔRGG was more shifted towards the lighter fractions (*Figure 4C,D*) and this was quantified (*Figure 4—figure supplement 1*). These findings suggest that RGG/RG motif of Aven is required for its recruitment to polysomes. To further investigate the role of RGG/RG motif methylation in polysomal localization of Aven, the FLAG-Aven was co-transfected with siPRMT1, which reduced PRMT1 expression by ~2.7-fold (*Figure 4—figure supplement 1*). Similar to what has been reported in fission yeast (*Bachand and Silver, 2004*), PRMT1 depletion did not have a major effect on the monosome/polysome ratio, thus, indicating that PRMT1 does not exert a major impact on global protein synthesis (*Figure 4—figure supplement 1*). Nonetheless, PRMT1 depletion shifted FLAG-Aven into the lighter fractions, as compared to a control (compare *Figure 4E* and *Figure 4G*, quantified in *Figure 4—figure supplement 1*). We confirmed that Aven does not co-sediment with mRNPs other than polysomes by showing that puromycin, an aminonucleoside antibiotic that dissociates polysomes (*Blobel and Sabatini, 1971*), leads to redistribution of both FLAG-Aven and rpS6 towards the lighter fractions corresponding to free ribosomal subunits, monosomes, and cytoplasmic mRNPs (*Figure 4F*). Taken together, our findings suggest that the arginine methylation of the Aven RGG/RG motif by PRMT1 is required for the association of Aven with polysomes.

TDRD3 and SMN are known to be polysome-bound (*Goulet et al., 2008*; *Sanchez et al., 2013*). Thus to determine their requirement for polysomal localization of Aven, FLAG-Aven was transfected in HEK293T cells depleted of SMN or TDRD3 or both using siRNAs. The depletion of TDRD3 (sixfold) and SMN (twofold) was confirmed by immunoblotting (compare *Figure 4G* and *Figure 4H*). FLAG-Aven co-sedimented in the polysome fractions with endogenous TDRD3, SMN, and rpS6 in siCTRL (*Figure 4G*, fractions 11 to 16). In contrast, depletion of both TDRD3 and SMN, but not single TDRD3 or SMN depletion (data not shown) reduced polysomal association of Aven, as compared to the control (*Figure 4H*). These findings suggest that TDRD3 and SMN are required for the recruitment of Aven to polysomes, whereby these proteins likely play a redundant role.

## Aven RGG/RG motif binds the G4 sequences of MLL1 and MLL4

Next, we reasoned that since RGG/RG motifs are encoded by G-rich sequences (codons: Gly GGN; Arg CGN or AGA/G), it is likely that certain mRNAs encoding RGG/RG motif-containing proteins may occasionally comprise a PG4 sequences (*Figure 5A*). To identify these RGG/RG encoding sequences and to identify all PG4 sequences in the coding sequences, we performed a bioinformatic search for PG4 sequences in mRNA-coding regions ($G_x$-$N_{1-7}$-$G_x$-$N_{1-7}$-$G_x$-$N_{1-7}$-$G_x$, where $x \geq 3$ and N is any of the nucleotides [A, C, G, or U]), and it revealed ~1600 PG4 in human ORFs (*Figure 5—source data 1*). We also provide a cG/cC score (*Figure 5—source data 1*), where >2 predicts higher G4 formation over Watson–Crick base pairing with neighboring sequences (*Beaudoin et al., 2014*). In addition, we also use RNAfold v2.1.7, which (*Figure 5—source data 1*) is a new scoring system to identify RNA G4-folding (*Lorenz et al., 2013*). RNAfold v2.1.7 was less efficient in predicting G4 formation than the cG/cC ratio (*Beaudoin et al., 2014*).

We observed a preference for amino acids G >> R/A/P > E/L consistent with the frequency of guanine in each codon. Some PG4 sequences identified encoded RG-rich sequences, as well as

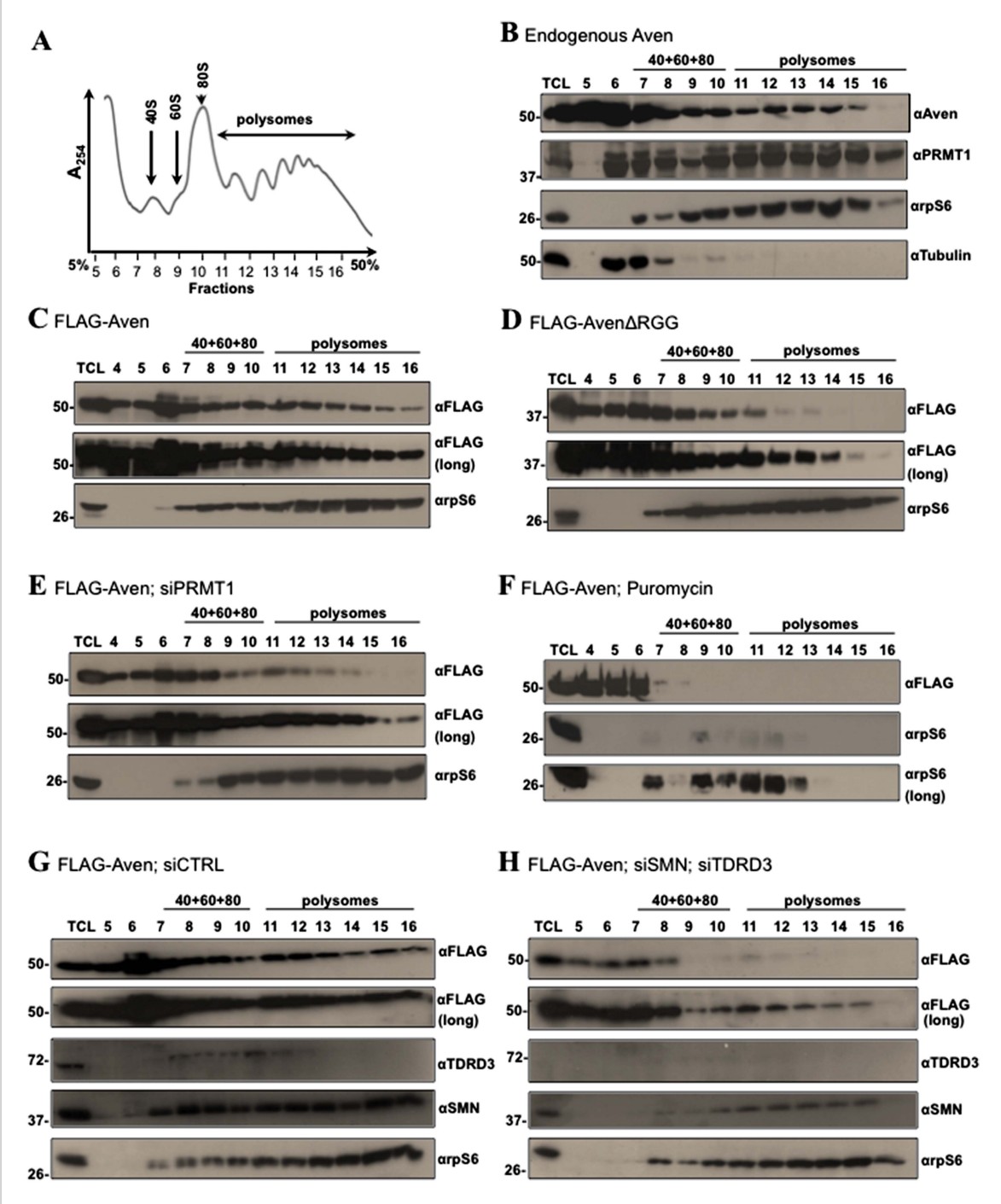

**Figure 4**. Methylation of Aven and its association with TDRD3 and SMN is required for polysomal localization. (**A**) Cytoplasmic extracts from HEK293T cells were sedimented by centrifugation on a 5–50% sucrose gradient. Polysome profiles were obtained by continuous monitoring of UV absorbance at 254 nm. 40S, 60S, and 80S indicate the positions of the respective ribosomal subunits and the monomer on the gradient. (**B**) The distribution of endogenous Aven and PRMT1 across the gradient of panel **A** was monitored by immunoblotting. Ribosomal protein rpS6 was used as a loading control, whereas β-tubulin served as a cytoplasmic marker. (**C–H**) The distribution of FLAG-Aven or FLAG-AvenΔRGG across the gradient was monitored by immunoblotting as well as FLAG-Aven in siControl, siPRMT1, siSMNsiTDRD3, or with puromycin treatment. Both short (5 s, panels **C–H**) and long exposures (30 s, panels **C–H**) are shown. rpS6 was used as a loading control. The exposure time was determined using a standard curve with increasing amounts of lysates expressing FLAG-Aven immunoblotted with anti-FLAG antibodies for various times. Each polysomal profile experiment was performed independently three times.

*Figure 4. continued on next page*

*Figure 4. Continued*

The following source data and figure supplement are available for figure 4:

**Source data 1**. Quantitative mass spectrometry of Aven and AvenΔRGG interacting proteins.
**Figure supplement 1**. Polysomal profiles of siRNA-treated cells and quantification of FLAG-Aven and FLAG-AVENΔRGG in polysomal fractions.

glycine-rich sequences. As Aven is a known survival protein and its depletion decreases the proliferation of leukemic cells (*Eismann et al., 2013*), we searched for mRNAs harboring PG4 sequences involved in leukemic cell survival. We identified human KMT2A, KMT2B, and KMT2D (*Figure 5—source data 1*), the family of MLL histone methyltransferases known to be mutated and/or part of fusion proteins, as the result of chromosomal translocations in leukemia (*Liedtke and Cleary, 2009*; *Smith et al., 2011*). KMT2A (MLL1) has a PG4 sequence (nucleotide 223–253) that encodes a glycine-rich sequence (AGSSGAGVPGG, *Figure 5B*; cG/cC score of 2.975, *Figure 5—source data 1*), KMT2B (MLL4) has 3 PG4 sequences (nucleotides 262–287; 3139–3166; 6274–6301) that encode an RGG/RG motif (RVQRGRGRG, *Figure 5B*; cG/cC score of 2.796, *Figure 5—source data 1*), a glycine-rich (RGAGAGGPRE; cG/cC score of 1.887, *Figure 5—source data 1*), an alanine-glycine-rich (RAGVLGAAGD; cG/cC score of 2.468, *Figure 5—source data 1*) sequences, and KMT2D has a PG4 from nucleotide 961 to 986 that encodes RVCRACGAG (cG/cC score of 2.029, *Figure 5—source data 1*). We examined whether Aven associated with the conserved PG4 RNA sequences (*Figure 5—figure supplement 1*) of KMT2A (MLL1) and KMT2B (MLL4) near the initiator ATG (i.e., 223–253 and 262–287). Indeed, Aven bound either MLL1 or MLL4 PG4 RNA sequences in the RGG/RG motif-dependent manner, but not to those harboring guanine to adenine mutants (*Figure 5C–F*). We next examined whether FLAG-Aven binds the endogenous MLL1 and MLL4 PG4 sequences in vivo using photocrosslinking with 4-thiouridine followed by immunoprecipitations (*Cahill et al., 2002*) and treatment with RNase I to digest RNAs into fragments of 50–300 nucleotides in length (*Huppertz et al., 2014*). HEK293T transfected with pcDNA3.1, FLAG-Aven, or FLAG-AvenΔRGG were prepared for CLIP, as described in 'Materials and methods' and IP with anti-FLAG antibodies (*Figure 5G*). Anti-FLAG immunoprecipitations of FLAG-Aven expressing cells, but not pcDNA3.1 or FLAG-AvenΔRGG-transfected cells, enriched ~30-fold for the MLL4 G4 sequence, whereas an RNA region 300 nucleotides downstream was not enriched (noG4; *Figure 5G*, right panel). MLL1 G4 sequence was also enriched (~fivefold) in FLAG-Aven immunoprecipitations, but not a region without G4 motifs (noG4) (*Figure 5G*, left panel). We investigated whether endogenous Aven could associate with the PG4s of MLL1 and MLL4 mRNAs. Photocrosslinking immunoprecipitations assays confirmed that MLL1 and MLL4 PG4 sequences associated with IP endogenous Aven, but not immunoglobulin G control albeit with a lower affinity (~fourfold to sixfold, *Figure 5H*) than with overexpressing FLAG-Aven (*Figure 5G*) and that ultraviolet light was required for this association (*Figure 5H*). The fold induction observed between Aven and the G4 sequences is likely under represented due to the fact that the crosslinks impede the reverse transcription reactions (*Huppertz et al., 2014*). These findings demonstrate that Aven is associated in vivo with the PG4s of MLL1 and MLL4.

We next performed in-line probing experiments, to determine whether the MLL PG4 sequences formed *bona fide* G4 structures. This assay compares the cleavage pattern in two conditions: in presence of $K^+$, which support G4 formation, and in presence of $Li^+$, which does not. G4 folding leads to an increased cleavage for the nucleotides within the loop regions, since they bulge out (*Beaudoin and Perreault, 2010*, *2013*). Using MLL1 and MLL4 PG4, we observed increased cleavage in the predicted loops of the formed G4, when incubated in $K^+$ compared to $Li^+$. However, no such difference was observed for a mutant RNA, where guanines were replaced with adenines (*Figure 5I*), confirming that the PG4 of MLL1 and MLL4 form G4 RNA structures.

## Aven regulates the protein, but not the mRNA, levels of MLL1 and MLL4 in T-ALL cells

Aven is required for the proliferation of T-ALL cells (*Eismann et al., 2013*). We generated Aven-deficient T-ALL cells, MOLT4 and CCRF-CEM using a lentivirus that expresses an shRNA against Aven and we achieved >80% knockdown (*Figure 6A,B*). We next monitored the levels of MLL1 and MLL4 protein by immunoblotting and their mRNAs by RT-qPCR. Both MLL1 and MLL4 protein levels were

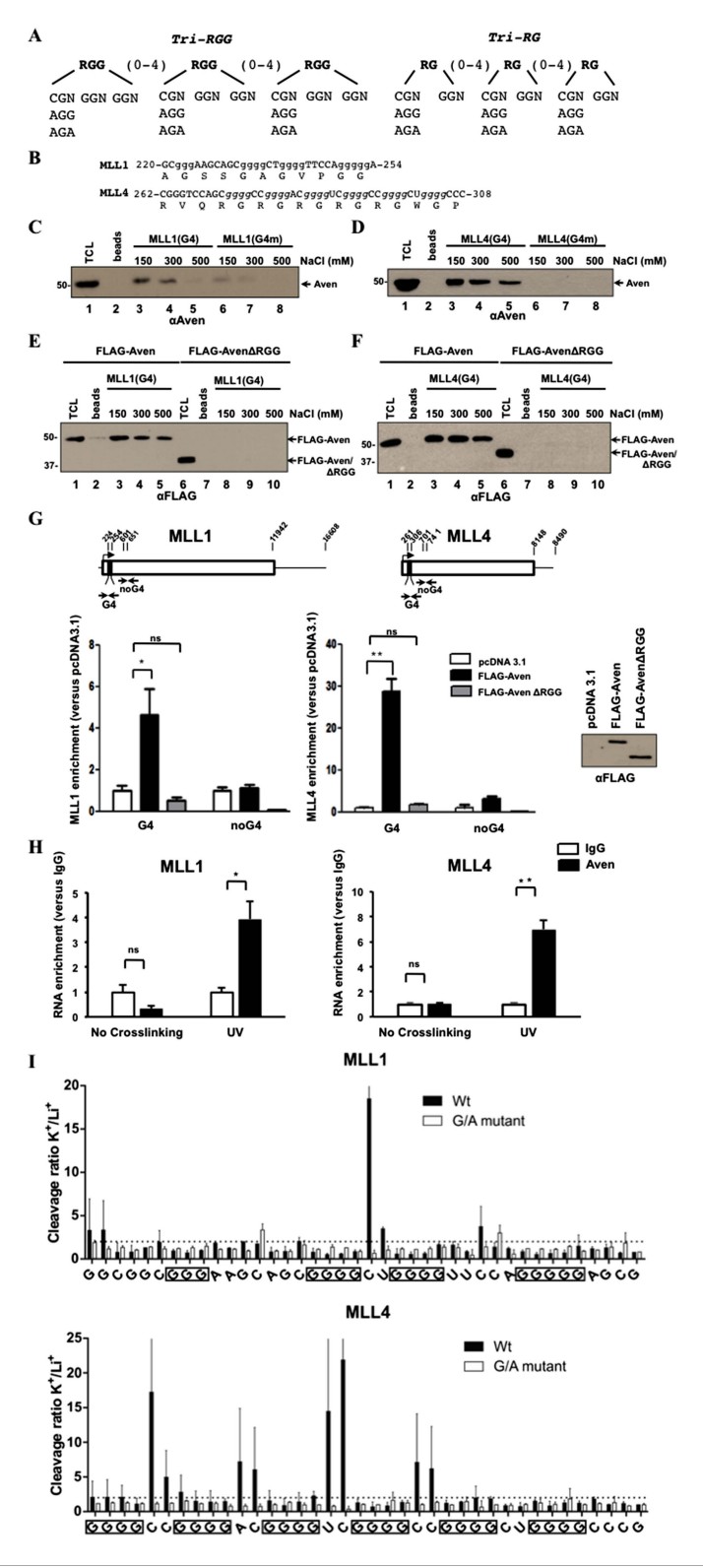

**Figure 5**. Aven RGG/RG motif binds G4 RNA structures of MLL1 and MLL4. (**A**, **B**) RNA sequences of the RGG/RG motifs and the PG4 motifs of MLL1 and MLL4. (**C**, **D**) Biotinylated MLL1 G4 or a mutant sequence (G4m), biotinylated MLL4 G4 or a mutant sequence (G4m) bound to Streptavidin beads were incubated with HEK293T cell lysates. The bound proteins were washed with increasing concentrations of NaCl and visualized by SDS-PAGE followed by *Figure 5. continued on next page*

*Figure 5. Continued*

immunoblotting with anti-Aven antibodies. (**E**, **F**) HEK293T cells expressing FLAG-Aven and FLAG-AvenΔRGG were processed as in panel **C**, **D** except the bound proteins were visualized by immunoblotting with anti-FLAG antibodies. (**G**) Photocrosslinking IP experiments were performed using anti-FLAG antibodies. The bound RNA was analyzed in triplicate from two biological replicates by RT-qPCR with the primers spanning the PG4 sequence or a sequence ~300 nucleotides downstream. The levels of bound RNA in immunoprecipitates were normalized to the levels of the total RNA in the input. Mean values are expressed as fold enrichment over pcDNA3.1. Error bars represent ±SEM. *p < 0.05, **p < 0.001, n.s. non-significant. The experiment was performed twice. (**H**) Photocrosslinking IP experiments were performed on HEK293T cells using anti-Aven antibodies. The bound RNA was analyzed in triplicates by real-time RT-PCR with the primers spanning the PG4 sequence, as indicated in panel **G**. The level of bound RNA in immunoprecipitates was normalized to the levels of the total RNA in the input. Mean values are expressed as fold enrichment over IgG. Error bars represent ±SEM. *p < 0.05, **p < 0.001, n.s. non-significant. The experiment was performed twice. (**I**) In-line probing of MLL1 and MLL4 PG4. The nucleotide sequence of the mixed lineage leukemias (MLLs) PG4 is shown below, the boxed guanines represent the predicted G-tracks. K⁺/Li⁺ ratios of the band intensities of the MLLs G4 (black) and G/A-mutant (white) for each nucleotide are shown. Error bars represent ± standard deviation. The experiment was performed twice. The dashed line represents a twofold change, an arbitrary set threshold that indicates G4 formation when exceeded.

The following source data and figure supplement are available for figure 5:

**Source data 1**. G4 sequences in coding regions of mRNAs.

**Figure supplement 1**. Sequence conservation of the MLL1 and MLL4 PG4 sequences.

reduced in Aven-deficient MOLT4 and CCRF-CEM cells (*Figure 6A,B*), whereas the levels of corresponding mRNAs remained unchanged (*Figure 6C,D*). Next, we investigated whether the reduced protein levels of MLL1 and MLL4 in Aven-depleted cells are associated with the reduced expression of *HOX* genes, which are well-established transcriptional targets of MLLs (*Krivtsov and Armstrong, 2007*). Aven-deficient cells exhibited reduced expression of several key *HOX* genes such as *HOXA9*, *HOXA7*, *HOXA1*, and *MEIS1* (*Figure 6E,F*). These findings suggest that Aven regulates the translation of MLL1 and MLL4 mRNAs thereby leading to an increase in MLL1 and MLL4 protein levels and an increase in the transcription of leukemic genes.

Aven-deficient MOLT4 and CCRF-CEM had decreased proliferation rates consistent with Aven being a survival protein (*Figure 6G,H*). Interestingly, the depletion of PRMT1 with shRNAs phenocopied Aven depletion, that is, also had reduced proliferation rates (*Figure 6G,H*). These findings show that Aven and PRMT1 are required for the proliferation of T-ALL cells.

## Aven and PRMT1 regulate the polysomal association of MLL1 and MLL4 mRNAs

We generated Aven⁻/⁻ HEK293T cells (clone #2) using CRISPR/Cas9 technology (*Mali et al., 2013*) and this was confirmed by immunoblotting (*Figure 7A*). A minor Aven ~34 kDa fragment (*Figure 7A*, denoted by asterisk) was observed in HEK293T cells and likely represents a Cathepsin D cleaved fragment reported previously (*Melzer et al., 2012*). Aven depletion did not influence the number of ribosomes involved in polysomes, as compared to control (*Figure 7B*), suggesting that general mRNA translation was not affected by Aven. However, a specific subset of mRNAs may still be particularly sensitive to changes in Aven levels. To investigate this possibility, we monitored the distribution of MLL1, MLL4, and β-actin mRNA in polysomal fractions from Aven-proficient and Aven-deficient cells. Polysomal fractions were isolated and MLL1, MLL4 mRNAs, as well as β-actin mRNA, were quantified by RT-qPCR. Loss of Aven expression reduced the amounts of MLL1 mRNA in the heavy polysomal fraction (fractions 12–15), with concomitant increase in lighter polysomal fractions (fractions 6 and 7; *Figure 7C*). Similarly, MLL4 mRNA was reduced in the heavy polysomal fraction (fractions 12–14) and a shift towards the light polysomal and pre-polysomal (fractions 7–8 and 10–11) (fractions 7–8). As a control, we monitored the levels of mRNAs encoding β-actin and depletion of Aven did not have a major effect on its distribution, inasmuch as the most of β-actin mRNA was associated with heavy polysomes (*Figure 7E*). These findings suggest that Aven selectively regulates the polysomal association of MLL1 and MLL4 mRNAs.

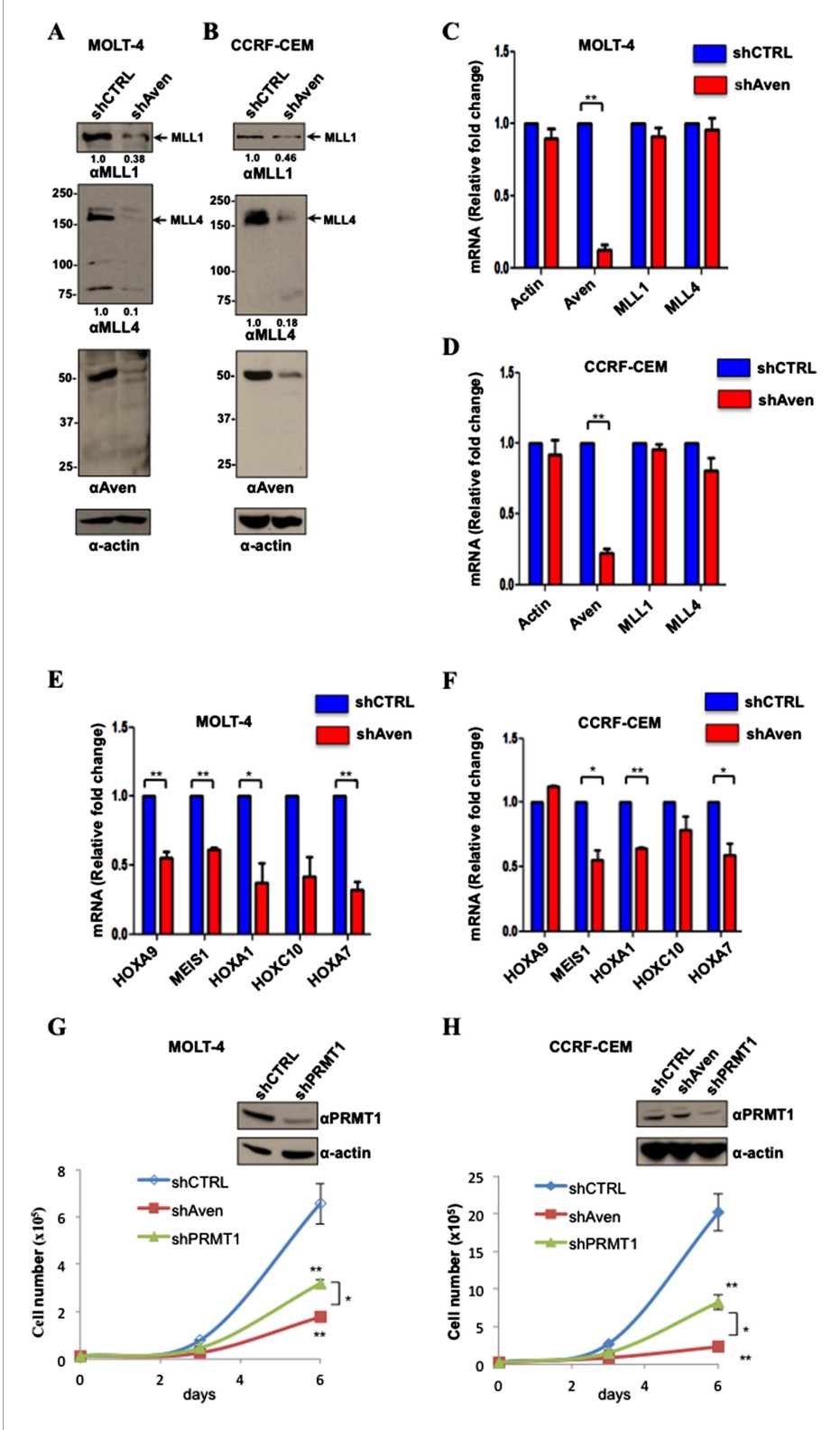

**Figure 6**. Aven regulates MLL1 and MLL4 protein expression required for leukemic cell survival. (**A**, **B**) Cellular lysates from MOLT-4 and CCRF-CEM cells stabling expressing shCTRL or shAven were separated by SDS-PAGE and immunoblotted with anti-MLL1, anti-MLL4, anti-Aven, and β-actin antibodies. The experiment was performed three times. (**C–F**) RT-qPCR of the indicated mRNAs was performed from RNA isolated from shCTRL and shAven MOLT-4

*Figure 6. Continued*

and CCRF-CEM cells and expressed as a relative fold change normalized to rpS6 levels. Error bars ±SEM is shown. The significance was measured by the Student's *t*-test and defined as *p < 0.05, **p < 0.001, n = 3. (**G**, **H**) Proliferation curves for shControl (CTRL), shPRMT1, and shAven MOLT-4 and CCRF-CEM cells are shown. Immunoblots confirm the depletion of PRMT1 in MOLT-4 and CCRF-CEM cells. Error bars ± standard deviation is shown. The data were analyzed using ANOVA (Analysis of Variance) followed by post hoc comparison using Tukey test. *p < 0.05, **p < 0.001, n = 3.

Since PRMT1 regulates the ability of Aven to associate with heavy polysomes, we investigated whether PRMT1-depleted cells also had reduced MLL1 and MLL4 mRNAs in heavy polysome fractions. Similarly to Aven, PRMT1 depletion (*Figure 8A*) did not influence the number of ribosomes involved in polysomes, as compared to a control, suggesting that PRMT1 does not affect global mRNA translation (*Figure 8B*). HEK293T cells were transfected with siGFP (siCTRL) or siPRMT1 for 72 hr, and the distribution of MLL1 and MLL4 mRNAs in polysomal fractions was monitored by RT-qPCR. Comparably to Aven, PRMT1 depletion had a striking effect on the distribution of MLL1 and MLL4 mRNAs in polysomes, as illustrated by their dramatic shift towards ligher fractions as compared to a control (*Figure 8C,D*). In contrast, depletion of PRMT1 resulted in a modest shift in β-actin mRNA, whereby the majority of β-actin mRNA remained in heavy polysome fractions. (*Figure 8E*, fractions 11–15). These findings suggest that Aven arginine methylation by PRMT1 regulates polysomal association of MLL1 and MLL4, but not β-actin mRNAs.

## DHX36 RNA helicase is required for Aven-dependent translation of RNAs

We closely examined the SILAC data for non-ribosomal proteins that are enriched in FLAG-Aven, but not with FLAG-AvenΔRGG immunoprecipitates that may function in unwinding G4 structures. The ATP-dependent RNA helicase DHX36, also known as G4 resolvase I, had an M/H ratio of 2.69 (*Figure 4—source data 1*). DHX36 has been reported to unwind G4 RNA structures (*Creacy et al., 2008*; *Lattmann et al., 2010*; *Booy et al., 2012*). We postulated that Aven could recruit the RNA helicases DHX36 to resolve G4 structures to facilitate protein synthesis. First, to validate whether the Aven RGG/RG motif is essential for interaction with the RNA helicase DHX36, we performed co-immunoprecipitations from cellular lysates expressing FLAG-Aven or FLAG-AvenΔRGG. The bound proteins were immunoblotted with anti-DHX36 antibodies. Indeed, DHX36 co-IP with FLAG-Aven, but not with FLAG-AvenΔRGG (*Figure 9A*). Moreover, we observed that DHX36 localized in the fast-sedimenting, heavier polysomal fractions with the control rpS6 (*Figure 9B*, fractions 12–15). To examine whether DHX36 influences the polysomal localization of MLL1 and MLL4 mRNAs, we monitored their mRNAs in polysomal fractions. Consistently with the observations for Aven and PRMT1, DHX36 depletion in HEK293T cells did not have a major effect on polysome absorbance profiles, thus, indicating that DHX36 does not affect global protein synthesis (*Figure 9C*). However, both MLL1 and MLL4, but not β-actin mRNA, shifted toward lighter polysomal fractions in cells depleted of DHX36, as compared to the control (*Figure 9D–F*). These findings suggest that similarly to PRMT1 and Aven, DHX36 regulates translation of MLL1 and MLL4 mRNAs.

## MLL4 G4 structure requires the Aven RGG/RG motif and PRMT1 for optimal translation

We next determined whether the polysomal association of MLL4 mediated by Aven requires an intact G4 structure. Reporter mRNAs harboring the G4 of MLL4 or a mutated G4 motif was inserted in-frame with the ORF of luciferase (*Figure 10A*). We examined whether Aven could promote the translation of the luciferase reporter protein in a G4-dependent manner and whether this was rescued by Aven re-expression in Aven-deficient cells. pGL3, pGL3-MLL4, or pGL3-MLL4 G4 mutant were transfected in Aven−/− HEK293T cells with pRenilla, as a control for transfection efficiency, along with either pcDNA3.1, FLAG-Aven, FLAG-AvenΔRGG, or FLAG-AvenR-K. The FLAG-AvenR-K protein was generated where the arginines in the RGG/RG motif were substituted for lysines to maintain the charge of the N-terminus of Aven. 24 hr post-transfection, the cells were harvested for dual luciferase assay. The presence of the MLL4 G4 sequence inhibited the relative luciferase activity by >75% in

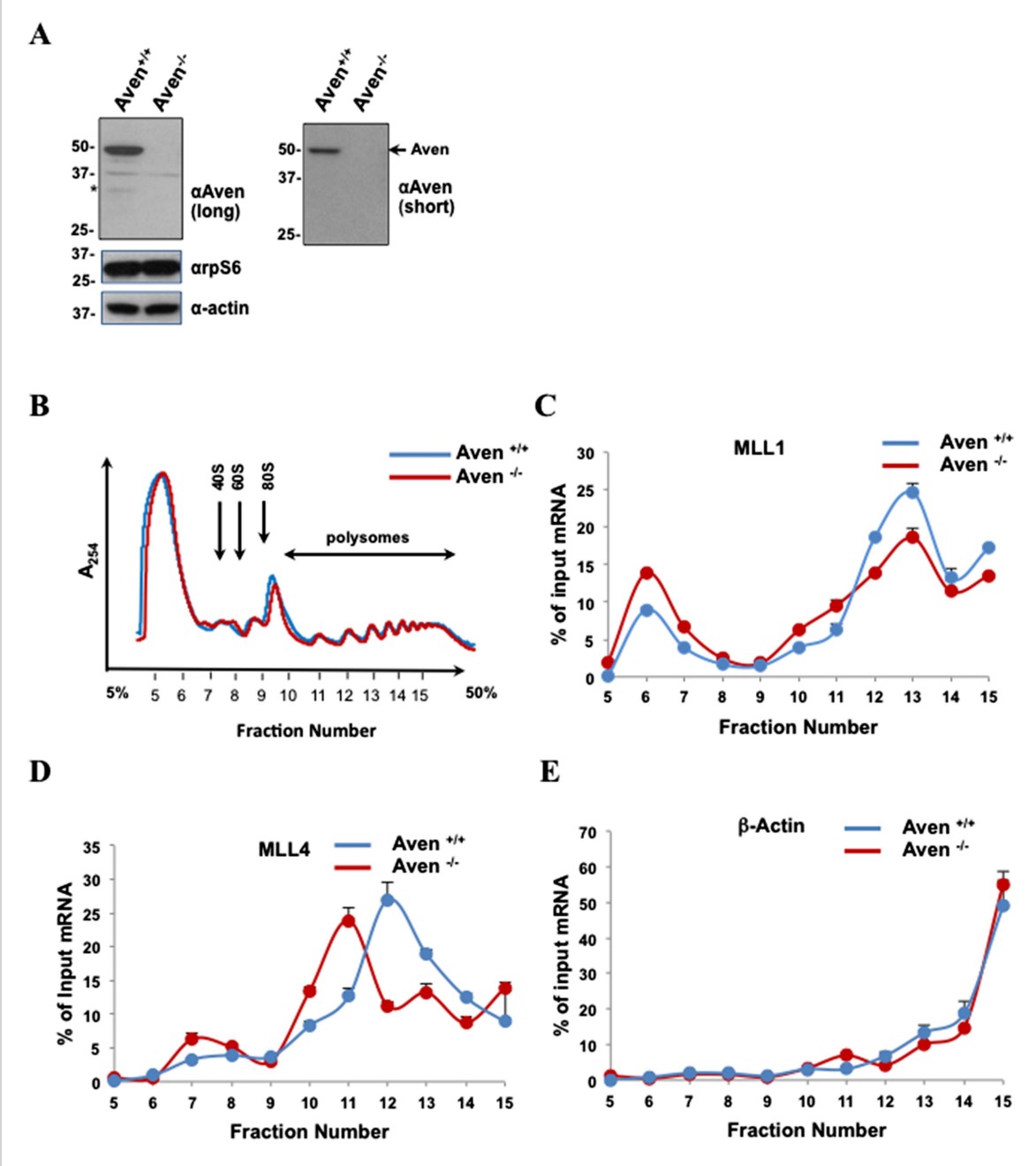

**Figure 7**. Aven regulates polysomal association of MLL1 and MLL4, but not β-actin mRNA. (**A**) Aven-deficient HEK293T cells were generated by CRISPR/Cas9. Stable clones were obtained Aven[+/+] (clone #7) and Aven[−/−] (clone #2). Anti-Aven, anti-rpS6, and anti-β-actin immunoblots of TCL are shown. The asterisks denote a minor Aven species of lower molecular mass. The band at ~37 kDa is a non-specific band recognized by the anti-Aven antibody. n = 3. (**B**) Polysome profiles of Aven[+/+] and Aven[−/−] HEK293T cells are shown. Cytoplasmic extracts from HEK293T cells were sedimented by centrifugation on a 5–50% sucrose gradient, shown as fraction numbers 5–15. Polysome profiles were obtained by continuous monitoring of UV absorbance at 254 nm. 40S, 60S, and 80S indicate the positions of the respective ribosomal subunits and the monomer on the gradient. (**C–E**) The indicated polysomal fractions were isolated, the RNA purified, and the presence of MLL1, MLL4, or β-actin was quantified by qRT-PCR. mRNAs in each fraction are represented as the percentage of input. Error bars represent ±SEM, n = 5.

Aven[−/−] HEK293T (*Figure 10B*, compare white and black bars labeled pcDNA3.1). The inhibition caused by the presence of the MLL4 G4 sequence was relieved by the transfection of FLAG-Aven, but not FLAG-AvenΔRGG nor FLAG-AvenR-K (*Figure 10B*, black bars). The presence of Aven did not

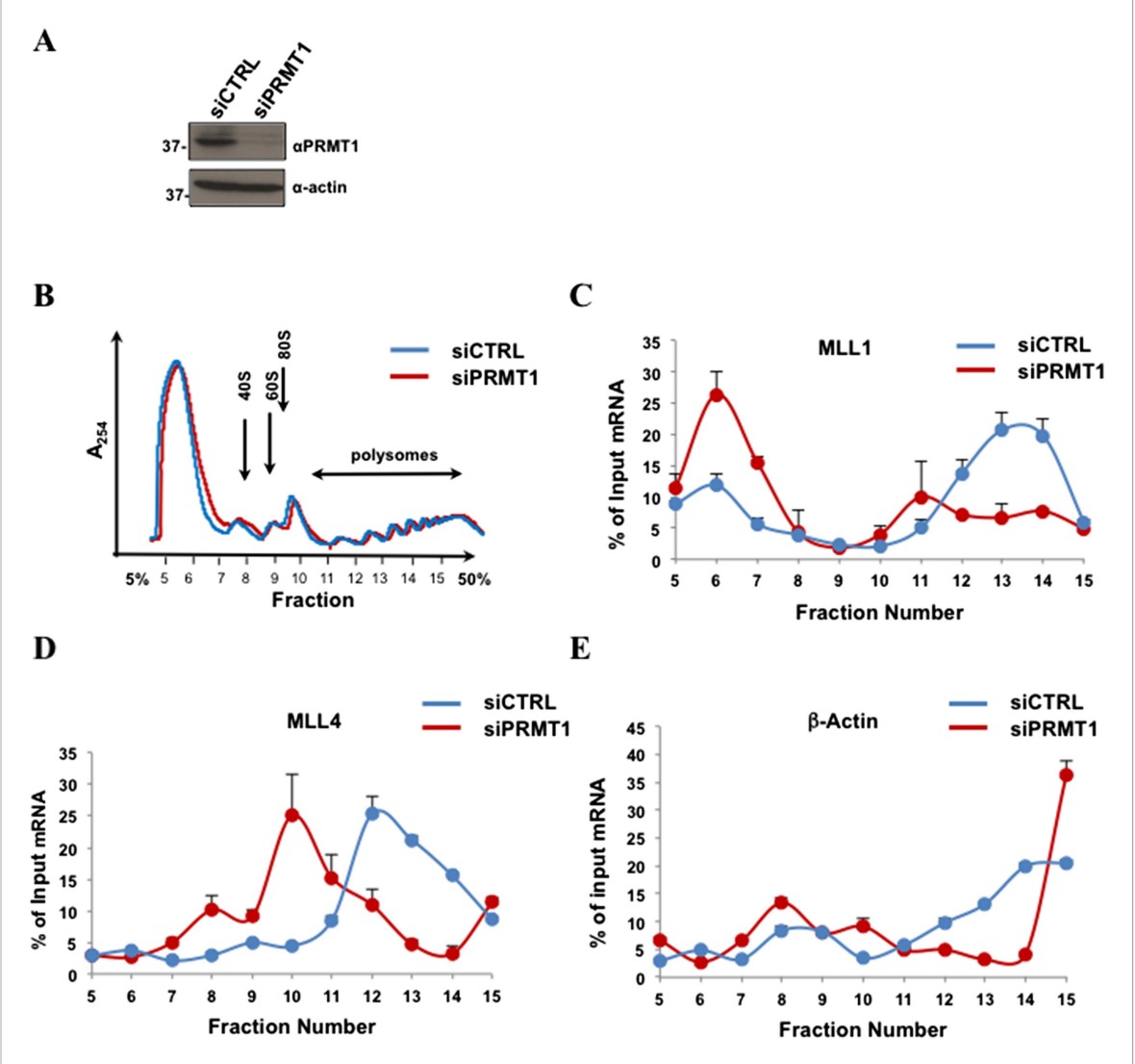

**Figure 8**. PRMT1 is required for the polysomal association of MLL1 and MLL4, but not β-actin mRNA. (**A**) PRMT1 was depleted by siRNA and cell extracts were immunoblotted with anti-PRMT1 or anti-β-actin antibodies. (**B**) Polysome profiles of siGFP (siCTRL) or siPRMT1 HEK293T cells are shown. Cytoplasmic extracts from HEK293T cells were sedimented by centrifugation on a 5–50% sucrose gradient, shown as fraction numbers 5–15. Polysome profiles were obtained by continuous monitoring of UV absorbance at 254 nm. 40S, 60S, and 80S indicate the positions of the respective ribosomal subunits and the monomer on the gradient. (**C**–**E**) The indicated polysomal fractions were isolated, the RNA purified, and the presence of MLL1, MLL4, or β-actin was quantified by qRT-PCR. mRNAs in each fraction are represented as the percentage of input. Error bars represent ±SEM, n = 2.

have any significant effects on the luciferase expressed from pGL3 or pGL3-MLL4:G4mutant (*Figure 10B*). We next examined arginine methylation by PRMT1 and DHX36 were required for the stimulation of translation by FLAG-Aven. HEK293T cells were transfected with FLAG-Aven and pGL3-MLL4:G4 in the presence of siCTRL, siPRMT1, or siDHX36. The absence of PRMT1 or DHX36 blocked FLAG-Aven from stimulating translation (*Figure 10C,D*), suggesting that both arginine methylation of Aven and DHX36 helicase activity are required to regulate the translation of mRNAs with G4 structures within their ORFs.

## Discussion

The *MLL* proto-oncogene encodes a histone methyltransferase implicated in epigenetic modifications, regulating gene expression for embryonic development and hematopoiesis (*Liedtke and Cleary, 2009*;

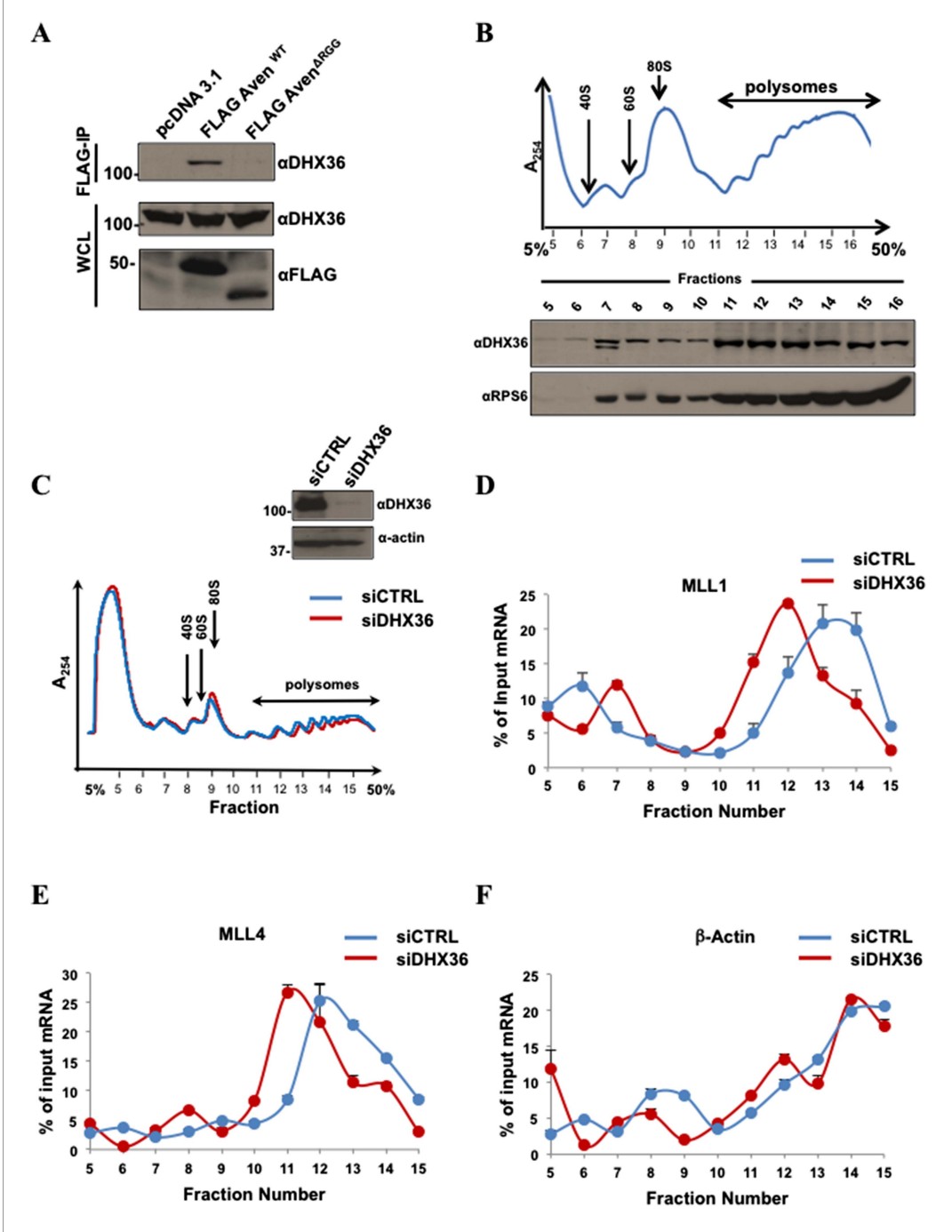

**Figure 9**. DHX36 is required for MLL1 and MLL4 mRNA polysomal association. (**A**) HEK293T cells expressing FLAG-Aven and the FLAG-AvenΔRGG were IP with anti-FLAG agarose beads and the bound proteins were immunoblotted with anti-DHX36 antibodies. TCL were immunoblotted with anti-DHX36 and anti-FLAG antibodies as indicated. (**B**) Proteins from the polysomal fractions isolated from HEK293T cells were TCA precipitated, separated by SDS-PAGE, and immunoblotted with anti-DHX36 and anti-rpS6 antibodies. The experiment was performed n = 4 times and a typical polysomal profile is shown. (**C**) Immunoblots of TCL from siGFP (CTRL) and siDHX36-transfected HEK293T cells are shown. Polysome profiles siCTRL and siDHX36-transfected HEK293T cells. Cytoplasmic extracts from HEK293T cells were sedimented by centrifugation on a 5–50% sucrose gradient, shown as fraction numbers 5 to 15. Polysome profiles were obtained by continuous monitoring of UV absorbance at 254 nm. 40S, 60S, and 80S indicate the positions of the respective ribosomal subunits and the monomer on the gradient. (**D–F**) The indicated polysomal fractions were isolated, total RNA isolated, and the presence of MLL1, MLL4, or

*Figure 9. continued on next page*

*Figure 9. Continued*

β-actin was quantified by RT-qPCR. mRNAs in each fraction are represented as the percentage of input. Error bars represent ±SEM, n = 2.

*Smith et al., 2011*). *MLL* is a recurrent site of DNA translocations resulting in an MLL fusion protein where the N-terminus of the *MLL* is fused to a variety of proteins (*Liedtke and Cleary, 2009*; *Smith et al., 2011*). In the present manuscript, we identify, within the mRNA-coding regions of MLL1 and MLL4, RNA elements that regulate its polysomal association and protein synthesis. These RNA elements are located between 200 and 300 nucleotides downstream of the initiator methionine ATG and encode protein sequences rich in glycines and arginine–glycine repeats in MLL1 and MLL4. The function of these N-terminal repeats is unknown. We show that Aven binds the MLL1 and MLL4 G4 RNA structures in vitro and in vivo with its RGG/RG motif. Aven was required for the translational regulation of MLL1 and MLL4, as Aven-deficient T-ALL cells exhibited decreased MLL1 and MLL4 protein expression and consequently decreased the expression of their downstream targets including, the *HOX* genes. The association of Aven with polysomes required the methylation of its RGG/RG motif by PRMT1 and interaction with methyl-binding proteins, TDRD3 and SMN. The Aven interaction with TDRD3 and SMN may require other protein or RNA components in the complex for enhanced association. Deficiency of Aven or PRMT1 in acute leukemic cell lines led to decreased cell proliferation. Taken together, our studies suggest that Aven regulates the translation of MLL1 and MLL4 required for cancer survival and that targeting this pathway may have therapeutic potential.

RGG/RG motifs have the biochemical properties to bind both RNA and proteins to fulfill their emerging roles in assembly of RNP complexes and translational control (*Rajyaguru and Parker, 2012*; *Thandapani et al., 2013*). The RGG/RG motif of yeast proteins Scd6, Npl3, and Sbp1 was shown to interact with the translational initiation factor eIF4G and repress translation by preventing the formation of pre-initiation complex (*Rajyaguru et al., 2012*). In trypanosomes, the RGG/RG motif of SCD6 is involved in regulating the type and number of RNP granules (*Krüger et al., 2013*). Amyloid-like fibers were formed when the RGG/RG motif of FUS was incubated with RNA (*Schwartz et al., 2013*). These fibers are characterized by the reversible transformation from soluble to polymeric state (*Han et al., 2012*; *Kato et al., 2012*). Although many proteins have an RGG/RG motif, the Aven RGG/RG motif may be more accessible, as it is located at the N-terminus and may protrude outwards. Since the Aven RGG/RG motif is not required for self-association (*Figure 2—figure supplement 1*), this suggests that an Aven dimer has 2 protruding RGG/RG motifs that can each mediate their own interactions. Therefore, we speculate that Aven functions as a scaffolding protein to assemble translationally competent RNPs for certain mRNAs containing G4 motifs (*Figure 11*).

DHX36 is a DEAH (aspartic acid, glutamic acid, alanine, histidine)-box helicase and it is the only G4 RNA resolvase known and is a major DNA G4 resolvase (*Creacy et al., 2008*; *Lattmann et al., 2010*). Aven associated with DHX36 to regulate translation of mRNAs with G4 structures. DHX36 knockdown increased the expression of PITX1 protein without changes in mRNA, suggesting that it functions in translational control (*Booy et al., 2014*). Ribosomal footprinting studies have led to the proposal that elongating ribosomes likely use accessory RNA helicases (*Rouskin et al., 2014*), and our data suggest that DHX36 may be such an accessory helicase. DHX36 null mice are embryonic lethal and deletion in the hematopoietic system using *Vav1-Cre* causes hemolytic anemia and defects at the proerythroblast stage with deregulation of genes with G4 motifs in their promoters (*Lai et al., 2012*), however, a role DHX36 in translational control was not examined.

Many PRMT1 substrates are RBPs with RGG/RG motifs (*Bedford and Clarke, 2009*) and some have been shown to associate with RNAs with G4 motifs such as Nucleolin, FUS, EWS, and FMRP (*Thandapani et al., 2013*). This suggests that several RBPs likely function in a similar manner to Aven in regulating accessibility of mRNPs with polysomes. It has been shown that the RGG/RG motif of FMRP is required for its polysomal association (*Blackwell et al., 2010*), however, whether arginine methylation by PRMT1 regulates association is unknown. Our findings show for the first time that arginine methylation by PRMT1 regulates translational control. It is known, however, that the yeast homolog of PRMT3 (RMT3) methylates rpS2, regulating the balance between the small and large

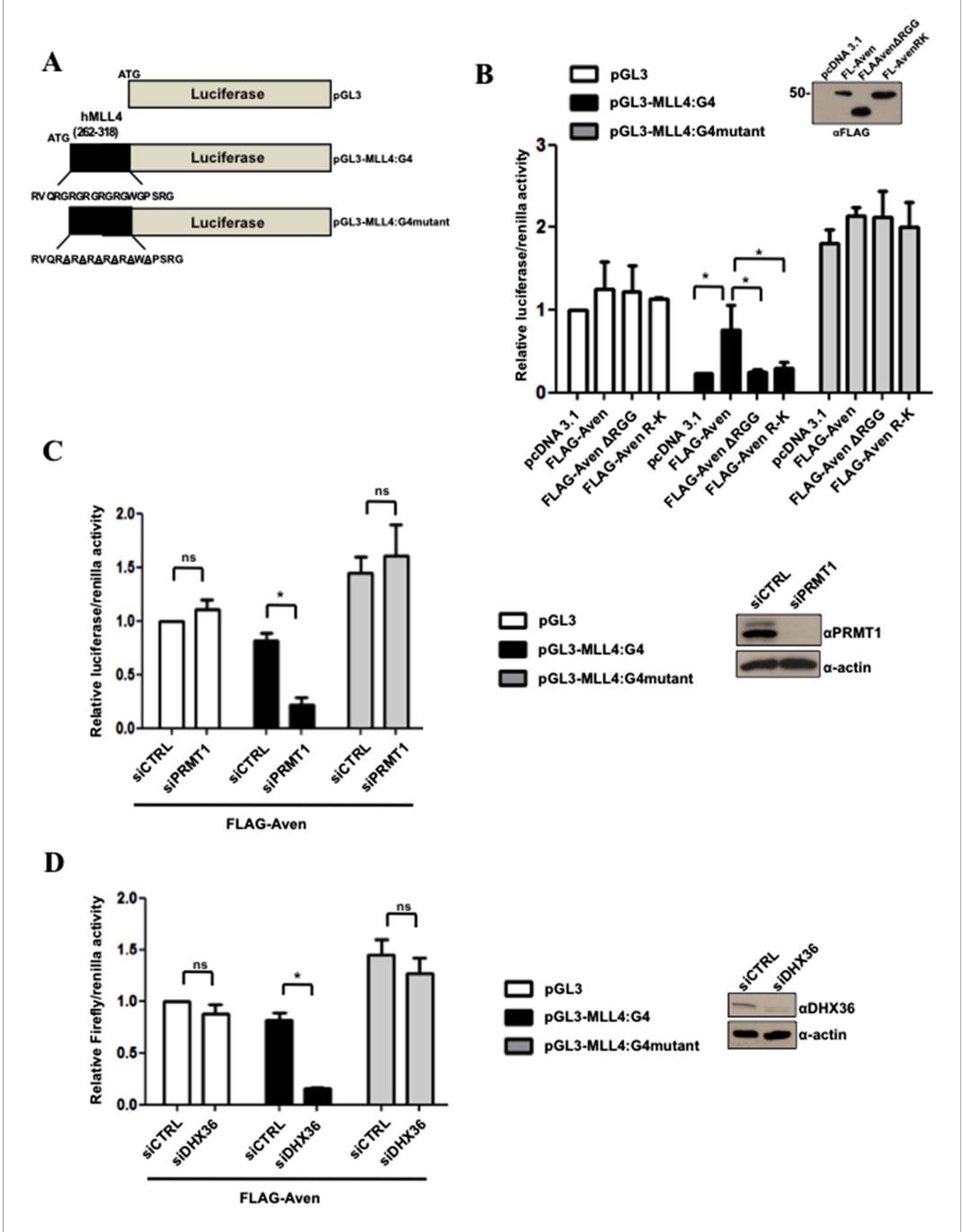

**Figure 10**. PRMT1 and Aven RGG/RG motif required for optimal translation of MLL4 G4 sequence. (**A**) Schematic of the luciferase reporter plasmid pGL3, as well as the chimeric pGL3-MLL4-G4 and pGL3-MLL4-G4mutant. pGL3-MLL4-G4 harbors the human MLL4 G4 sequence nucleotide 262 to 318 inserted in-frame at the N-terminus of luciferase, while pGL3-MLL4-G4mutant contains glycine to alanine mutations that disrupts the G4 structure.
(**B**) Aven$^{-/-}$ HEK293T cells were transfected with the following reporter genes pGL3, pGL3-MLL4-G4, or pGL3-MLL4-G4mutant and pRenilla as well as pcDNA3.1, FLAG-Aven, FLAG-AvenΔRGG, or FLAG-AvenR-K. The cells were harvested 24 hr post-transfection and dual luciferase assays were performed. The relative luciferase/Renilla ratio was normalized to 1.0 in pGL3 pcDNA3.1 transfected cells. Extracts were immunoblotted with anti-FLAG antibodies to confirm Aven, AvenΔRGG, or AvenR-K expression. Error bars represent standard deviation values. The experiments were performed three independent times (n = 3) and each independent experiment was performed in technical triplicates. The significance was measured by ANOVA followed by post hoc comparison using Tukey test. *p < 0.05.
*Figure 10. continued on next page*

*Figure 10. Continued*

(**C**) HEK293T cells were co-transfected with FLAG-Aven and either siGFP (siCTRL) or siPRMT1 along with the following reporter genes pGL3, pGL3-MLL4-G4, or pGL3-MLL4-G4mutant and pRenilla. The cells were harvested 24 hr post-transfection and dual luciferase assays were performed. The relative luciferase/Renilla ratio was normalized to 1.0 in pGL3 siCTRL transfected cells. Extracts were immunoblotted with anti-PRMT1 or anti-β-actin antibodies, as indicated. Error bars represent standard deviation values. The experiments were performed three independent times (n = 3) and each independent experiment was performed in technical triplicates. The significance was measured by ANOVA followed by post hoc comparison using Tukey test. *$p < 0.05$, n.s. non-significant. (**D**) HEK293T cells were co-transfected with FLAG-Aven and either siGFP (siCTRL) or siDHX36 along with the following reporter genes pGL3, pGL3-MLL4-G4, or pGL3-MLL4-G4mutant and pRenilla. The cells were harvested 24 hr post-transfection and dual luciferase assays were performed. The relative luciferase/Renilla ratio was normalized to 1.0 in pGL3 siCTRL transfected cells. Extracts were immunoblotted with anti-DHX36 or anti-β-actin antibodies, as indicated. The error bars represent ± the standard deviation. Experiments were performed three times (n = 3) and each experiment was analyzed in triplicates. The significance was measured by ANOVA followed by post hoc comparison using Tukey test. *$p < 0.05$, n.s. non-significant.

ribosomal subunits (*Bachand and Silver, 2004*). However, mammalian PRMT3 did not influence ribosomal assembly or polysomal formation (*Swiercz et al., 2004*).

It is known that secondary RNA structure including G-quadruplex structures within mRNAs hinder mRNA translation (*Koromilas et al., 1992*; *Sonenberg and Hinnebusch, 2009*). Stable RNA secondary structures within the 5′-UTRs of mRNAs reduce cap-dependent translation by preventing assembly of the translational initiation machinery at the 5′-cap and also impair the scanning of the start site AUG by the initiation complex (*Beaudoin and Perreault, 2010*; *Bugaut and Balasubramanian, 2012*). Secondary structure in the 5′ UTRs including G4 motifs has been shown to require eIF4A for optimal translation output (*Wolfe et al., 2014*). The 5′-UTR of NRAS and Zic-1, which harbor G4 structures, reduce translation of a reporter luciferase (*Kumari et al., 2007*; *Arora et al., 2008*). G-quadruplex structures within ORFs of the virally encoded EBNA1 transcript were shown to hinder translational elongation by either ribosomal pausing or ribosomal dissociation (*Murat et al., 2014*). We now extend these observations and identify a mechanism regulated by arginine methylation that leads to the positive regulation of mRNAs with G4 structures within their coding region.

Aven is overexpressed in acute leukemia and was proposed to be a prognostic factor in acute childhood lymphoblastic leukemia for poor outcome (*Choi et al., 2006*). Aven is a well-established cell survival protein or inhibitor of apoptosis that prevents apoptosis by stabilizing pro-survival protein Bcl-x$_L$ and inhibiting the function of pro-apoptotic protein Apaf-1 (*Chau et al., 2000*; *Kutuk et al., 2010*). It was reported that an N-terminal deleted fragment of Aven cleaved by cathepsin D harbors its anti-apoptotic function (*Melzer et al., 2012*), however, such a ∼30 kDa Aven species was not visible in MOLT4 and CCRF-CEM cells (*Figure 6A,B*) and was faintly observed in HEK293T (*Figure 7A*) as previously described (*Melzer et al., 2012*). Thus, cathepsin D mediated cleavage of

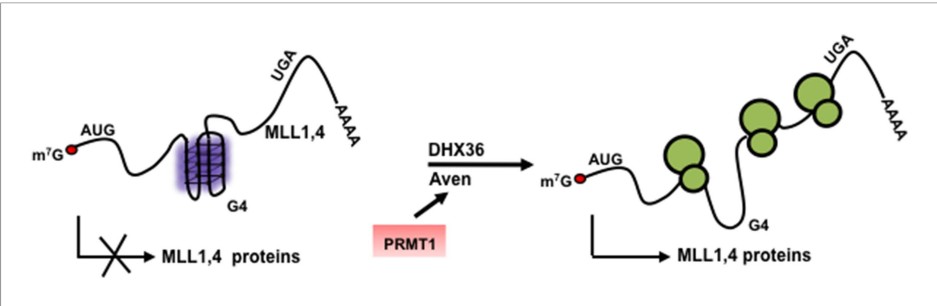

**Figure 11**. Model denoting the role of arginine methylated Aven by PRMT1 and DHX36 in the translation of G4 harboring MLL1 and MLL4 proteins.

Aven is unlikely involved in the regulation of translational control described herein. In addition to its pro-survival functions, Aven was identified to be essential for progression of acute leukemia in mice (*Eismann et al., 2013*). The regions required for association with Bcl-x$_L$ and Apaf-1 reside C-terminal of the RGG/RG motif. Taken together with our findings, this suggests that Aven uses several mechanisms to increase cell survival, (1) preventing apoptosis via Bcl-x$_L$ and Apaf-1, and (2) favoring the translation of mRNAs, including those encoding MLL1 and MLL4 required for cell survival.

PRMT1 was shown to be essential for MLL by the *MLL-EEN* gene fusion protein (*Cheung et al., 2007*). The *EEN* fusion partner leads to the recruitment of PRMT1 to methylate histones and lead to gene activation (*Cheung et al., 2007*). Our findings identify a new role for PRMT1 in the cytoplasm that is required for cancer cell survival. This pathway is amenable to therapeutic intervention with future PRMT1 inhibitors and specific RNA G-quadruplex ligands.

# Materials and methods

## Cells, reagents, and antibodies

HEK293T, U2OS, MOLT-4, and CCRF-CEM were from the American Type Culture Collection (Manassas, VA). $PRMT1^{FL/-;CreERT}$ MEFs were described previously (*Yu et al., 2009*). Protein A-Sepharose, 4-hydroxytamoxifen (OHT), anti-FLAG (M2) antibody-coupled agarose beads, mouse anti-FLAG (M2), anti-Myc, anti-MLL4 (WH0009757M2), and anti-α-tubulin were purchased from Sigma–Aldrich (St. Louis, MO). Mouse anti-Aven (ab77014, Abcam, UK) was used for immunoprecipitations, and rabbit anti-Aven (ProSci 2413, ProScience, San Diego, CA) was used for immunoblotting. Mouse anti-rpS6 was from Santa Cruz Biotechnology (Santa Cruz, CA). Rabbit anti-PRMT1 and ASYM25b antibodies were purchased from Millipore (Billerica, MA). Anti-SMN was from Transduction Laboratories (Lexingtong, KY). Anti-TDRD3 was a kind gift from Mark Bedford (Smithville, TX). Anti-MLL1 antibodies (A300-086A) were from Bethyl Laboratories (Montgomery, TX) and anti-DHX36 (ab70269) was from Abcam.

## DNA constructs

The full-length human FLAG-Aven and FLAG-AvenΔRGG lacking amino acids (1–73) were cloned in pcDNA3.1. The full-length human Aven was also cloned in pcDNA3.1 with 5 epitope tags of Myc at the N-terminus between the *Bam*HI and *Xho*I sites. GST Aven RGG/RG including amino acids (1–73) was cloned in pGEX5x-1 plasmid. FLAG-AvenR-K was assembled using G-blocks purchased from IDT. Aven arginines R5, R8, R11, R13, R14, R17, R19, R24, R28, R37, R50, R51, R53, R55, R57, R60, R63, and R66 were replaced with lysines. pGL3-MLL4:G4 and pGL3-MLL4mutant was generated by cloning MLL4 nucleotide sequence (262–318) (5′-CGG GTC CAG CGG GGC CGG GGA CGG GGT CGG GGC CGG GGC TGG GGC CCG AGT CGA GGC-3′) or (5′-CGC GTC CAG CGC GCC CGC GCC CGC GCC CGC GCC CGC GCC TGG GCC CCG AGT CGA GGC-3′) in fusion with the luciferase coding region in the pGL3 basic plasmid at *Sac*I and *Nco*I sites.

## Photocrosslinking and immunoprecipitations experiments

HEK293T cells expressing either pcDNA3.1, FLAG-Aven, and FLAG-AvenΔRGG were incubated with 100 μM 4-thiouridine (4SU) for 8 hr prior to crosslinking. The cells were washed with ice-cold PBS (phosphate buffered saline) and irradiated with 0.15 J/cm$^2$ of 365-nm UV light at 4°C. The cells were collected by centrifugation at 514×g for 1 min at 4°C. The cell pellets were resuspended in lysis buffer supplemented with protease inhibitors (Roche, Germany) and 0.5 U/ml RNasin (Promega, Madison, WI) (*Cahill et al., 2002*). 10 μl of 1:250 dilution of RNase I (Life Technologies, Carlsbad, CA) and 2 μl Turbo DNase (Life Technologies) were added to the lysate while shaking at 37°C for 3 min. The lysates were then cleared and IP with 25 μl anti-FLAG M2 affinity beads (Sigma). The beads were washed twice with high-salt buffer, twice with the lysis buffer and incubated with proteinase K buffer (containing 1.2 mg/ml Proteinase K) for 20 min at 37°C. RNA then was isolated through TRIzol reagent and subjected to RT-qPCR. For endogenous Aven, UV-crosslinked lysates were processed as described above expect they were incubated with rabbit IgG or anti-Aven (Proscience).

The reverse primers were used for the reverse transcription reaction: for MLL1 G4 structure, the RT reverse primer is 5′-GAG GAG GCT GCT GAG GCG GC-3′; for MLL1 negative control region (~300 bp

downstream of G4), the RT reverse primer is 5′-TCT TCT TGA TCT TAT CTC CA-3′; for MLL4 G4 structure, the RT reverse primer is 5′-CTC TCC TCC TCC GGC ACG CAG C-3′; for MLL4 negative control region (~300 bp downstream of G4), the RT reverse primer is 5′-GAT TGT CAC AGC TGC TTC TGC-3′ qPCR was performed with MLL1 (G4 structure, forward) 5′-GGC GGG AAG CAG CGG GGC TG-3′ and MLL1 (G4 structure, reverse) 5′-CTG AGG CGG CGG CCG CTC CC-3′; MLL1 (negative control, forward) 5′-CAT CTG TGT TTT CCC CTC TA-3′ and MLL1 (negative control, reverse) 5′-CTT ATC TCC AGA TTT GGT CT-3′; MLL4 (G4 structure, forward) 5′-GCG CCG GCT CCG CCG CCT GT-3′ and MLL4 (G4 structure, reverse) 5′-GCA CGC AGC CTC GAC TCG GG-3′; MLL4 (negative control, forward) 5′-TGC AGG AGG AAG CAG CAA GC-3′ and MLL4 (negative control, reverse) 5′-CTG CTT CTG CCA CCA CTA CT-3′.

## Polysome profiling

Polysome profiling has been performed as described in detail (*Gandin et al., 2014*). Briefly, HEK293T cells in 150-mm plates were transfected with the indicated expression plasmids using Lipofectamine 2000. Approximately 70% confluent cells were treated with 100 µg/ml cycloheximide for 5 min to 'freeze' mRNA translation. The cells were washed twice with ice cold-PBS and lysed in hypotonic lysis buffer containing 50 mM Tris-HCl (pH 7.5), 2.5 mM MgCl₂, 1.5 mM KCl, 100 µg/ml cycloheximide, 2 mM DTT, 0.5% Triton X-100, and 0.5% sodium deoxycholate. The lysates were spun at 13,000 rpm for 10 min at 4°C and layered onto a 5–50% sucrose gradient as previously described (*Gandin et al., 2013*). The gradients were formed using a SW40 rotor (Beckman, Pasadena, CA) at 36,000 rpm for 2 hr at 4°C. One ml fractions were collected by upward displacement with 60% sucrose and absorbance was continuously recorded at 254 nm using ISCO fractionator (Teledyne, ISCO). Collected fractions were precipitated with 10% TCA, separated by SDS-PAGE and proteins visualized by immunoblotting. For RNA analysis, 800 µl TRIZOL was added to the 1 ml fractions and RNA was isolated using standard procedures. Isolated RNA was quantified using RT-qPCR. The cDNA samples were serially diluted and the efficiency and *Cq* values were used to generate a standard curve (*Piques et al., 2009*). One standard curve was generated for each primer pair. All standard curves had $R^2$ value higher than 0.99, with a slope between −3.58 and −3.10. Each data point for each fraction was plotted against the standard curve to calculate the percentage of input.

## RNA binding assays

Biotinylated RNAs were purchased from IDT (Coralville, IA). The RNAs were dissolved in binding buffer (10 mM Tris-acetate, pH 7.7, 200 mM potassium acetate, 5 mM magnesium acetate), heated to 75°C for 10 min, and allowed to renature at 21°C for 5 min (*Phan et al., 2011*). For the RNA binding assays, 100 nM final concentration of biotinylated RNA was incubated with cellular lysates containing 2 mg/ml heparin for 1 hr on ice. Then, 25 µl of 50% Streptavidin agarose slurry was added and incubated at 4°C for 30 min with constant end-over-end mixing. The beads were then washed 2× with cell lysis buffer with increasing salt concentration and once with PBS. The samples were then boiled with 25 µl of 2× Laemmli buffer, resolved by SDS-PAGE and visualized by immunoblotting.

## Peptide RNA-binding assay

Fluorescein-labelled RNA was dissolved in binding buffer (10 mM Tris-acetate, pH 7.7, 200 mM potassium acetate, 5 mM magnesium acetate), heated to 75°C for 10 min, and allowed to renature at 21°C for 5 min (*Phan et al., 2011*). The biotinylated peptides (20 pmol in 50 µl ddH₂O) were allowed to bind the streptavidin-coated high-capacity binding plates (Pierce #15,503, Rockford, IL) overnight at 4°C or 2 hr at room temperature. The peptides were removed and the plates were washed four times with binding buffer. Different concentrations of the fluoresceinated RNA (IDT) was allowed to bind the peptides for 1 hr at room temperature. The unbound RNAs were removed and the plates were washed four times with binding buffer. Fluorescence was measured at 521 nm on a Synergy H4 instrument (BioTek, Winooski, VT). The peptides were purchased from Epicypher Inc and their sequence are Aven50-74 biotin-RRGRGRGRGFRGARGGRGGGGAPRG, termed DiRGG; Aven50-74(Me2a) biotin-RRGRGRGRGFRGAR(Me2a)GGR(Me2a)GGGGAPRG, termed DiRGGme; Aven2-26: biotin-QAERG ARGGRGRRPGRGRPGGDRHS, termed TriRG; Aven2-26(Me2a): biotin-QAER(Me2a)GAR(Me2a)GGR(Me2a) GRRPGR(Me2a)GRPGGDRHS, termed TriRGme.

## RT-qPCR primers

| Gene | Primer | Sequence (5′ → 3′) |
| --- | --- | --- |
| MLL1 | Forward | GAGGACCCCGGATTAAACAT |
| | Reverse | GGAGCAAGAGGTTCAGCATC |
| MLL4 | Forward | CAGACCCGGCAGACAGATGAG |
| | Reverse | AGATGTTACGTAGTCAAGGCACA |
| rpS6 | Forward | AATGGAAGGGTTATGTGGTCCG |
| | Reverse | CCCCTTACTCAGTAGCAGGC |
| HOXA9 | Forward | TCAAAAGGATAGCGCTGCCA |
| | Reverse | TGCATTACCAGAGAGCCGTG |
| MEIS1 | Forward | ACCGTTTGCGACTTGGTACT |
| | Reverse | TGCTCACAACCAGACAGCTC |
| Actin | Forward | ACCACACCTTCTACAATGAGC |
| | Reverse | GATAGCACAGCCTGGATAGC |
| HOXA1 | Forward | ACCAAGAAGCCTGTCGCTC |
| | Reverse | ACTTTCCCTGTTTTGGGAGGG |
| HOXC10 | Forward | ACCACAGGAAATTGGCTGAC |
| | Reverse | GATCCGATTCTCTCGGTTCA |
| Aven | Forward | CTCTGCCTCCGACTCAAC |
| | Reverse | CCTTGCCATCATCAGTTCTC |
| HOXA7 | Forward | AAGCTGCCGGACAACAAATC |
| | Reverse | GAAGCCCCCGCCGTATATTT |
| GAPDH | Forward | AATCCCATCACCATCTTCCA |
| | Reverse | TGAGTCCTTCCACGATACCA |

## Generation of stable clones

CCRF-CEM and MOLT-4 cells maintained in RPMI (Roswell Park Memorial Institute medium) with 10% FBS (Fetal Bovine Serum) were transduced with lentiviruses harboring either shRNA targeting AVEN (5′-CCG GGA GAA TGA TGA ACA GGG AAA TCT CGA ATT TCC CTG TTC ATC ATT CTC TTT TTT G-3′), PRMT1 (5′-CGG GTG TTC CAG TAT CTC TGA TTA CTC GAG TAA TCA GAG ATA CTG GAA CAC TTT TTG-3′), or control shRNA in the vector pLKO.1. The lentiviruses were generated in HEK293T cells following recommended manufacturer's protocol with modifications in transfection as follows (9 µg psPAX2-; 4 µg vsv-g; 9 µg shRNA) per 10-cm plate. The shRNAs were purchased from the shRNA library from Dharmacon (Lafayette, CO). Post-infection, bulk populations of stably infected cells were selected with 2 µg/ml puromycin.

## siRNA transfections

Small interfering RNAs (siRNAs; Dharmacon Inc.) were transfected using Lipofectamine RNAi MAX (Invitrogen, Carlsbad, CA) as per the manufacturer's protocol. The final concentration of the siRNA was 40 nM and the cells were lysed 72 hr post-transfection. The siRNA target sequence for PRMT1 was 5′-CGU CAA AGC CAA CAA GUU A-3′. The siRNA target sequences for Aven were siAven 5′-GAG GAG AAA GAA UGG GAU AUU-3′. For SMN, TDRD3, and DHX36 siRNAs, SMARTpools were purchased from Dharmacon Inc.

## Immunoprecipitations and immunoblotting

*PRMT1*$^{FL/-;CreERT}$ MEFs or HEK293T cells were transfected using Lipofectamine 2000 (Invitrogen) as per manufacturer's instructions. After 24 to 48 hr, the cells were lysed with cell lysis buffer (20 mM Tris pH 7.4, 150 mM NaCl, 1 mM EDTA (Ethylenediaminetetraacetic acid), 1 mM EGTA (Ethylene glycol tetraacetic acid), 1% Triton X-100). For immunoprecipitations, cell lysates were incubated with the

primary antibody for 1 hr on ice. Then, 25 µl of 50% protein A-Sepharose slurry was added and incubated at 4°C for 45 min with constant end-over-end mixing. The beads were then washed twice with cell lysis buffer and once with 1× PBS. The samples were then boiled with 25 µl of 2× Laemmli buffer, resolved by SDS-PAGE, transferred to nitrocellulose membranes and the proteins visualized by immunoblotting.

## Recombinant GST pull-down assays

U2OS cells transfected with FLAG-Aven full length or FLAG-AvenΔRGG were lysed in lysis buffer 48 hr after transfection. Cell lysates were prepared and incubated for 1 hr at 4°C with 20 µl of 50% slurry of the purified GST-Tudor proteins bound to the glutathione agarose. 1 µg of GST protein was used for each pull-down. Following the incubation, the beads were washed three times with lysis buffer and the proteins eluted in 1× Laemmli buffer. The bound proteins were analyzed by SDS-PAGE and visualized by immunoblotting.

## Luciferase reporter assay

Aven$^{-/-}$ cells seeded in 24-well plates were transfected with 500 ng of either pGL3 control, pGL3 pGL3-MLL4:G4, or pGL3-MLL4:G4mutant along with 1 µg of the different FLAG-Aven constructs and 5 ng pRenilla per well. 24 hr post-transfection, the cells were lysed with passive lysis buffer and the renilla and firefly luciferase activities were measured using Dual-luciferase reporter assay kit from Promega. Aven$^{-/-}$ cells were transfected with either siControl, siPRMT1, or siDHX36. 24 hr post-transfection, the cells were seeded in 24-well plates. The following day, the cells were transfected with 500 ng of either pGL3 control, pGL3-MLL4:G4, or pGL3-MLL4:G4mutant along with 1 µg FLAG-Aven and 5 ng pRenilla per well. 24 hr post-transfection, the cells were lysed with passive lysis buffer and the renilla and firefly luciferase activities were measured using the Dual-luciferase reporter assay kit.

## In-line probing

In-line probing assays were performed, as previously described (Beaudoin et al., 2013). Briefly, trace amounts of labelled RNA (50,000 cpm) were heated at 70°C for 5 min and then slow-cooled to room temperature over 1 hr in buffer containing 100 mM Tris-HCl (pH 7.5) and 100 mM LiCl or KCl of 10 µl. Following this incubation, the final volume of each sample was adjusted to 20 µl such that the final concentrations were 50 mM Tris-HCl (pH 7.5), 20 mM MgCl$_2$, and 100 mM LiCl or KCl. The reactions were then incubated for 40 hr at room temperature, ethanol-precipitated and the RNAs dissolved in formamide dye loading buffer (95% formamide, 10 mM EDTA, and 0.025% bromophenol blue). The radioactivity of the in-line probing samples was measured, and equal amounts in terms of desintegrations per minute of all conditions of each candidate were fractionated on denaturing (8 M urea) 10% polyacrylamide gels. The SAFA software was used to quantify each band. The intensity of the band incubated in KCl was then divided by the intensity of the corresponding band incubated with LiCl. G4 formation is confirmed when this value exceed 2 (Beaudoin and Perreault, 2010, 2013). Histograms show the mean result and standard deviation of two separate experiments, that is, two different RNA transcription, labeling, and in-line probing.

The sequences used were for wild-type MLL1 5′-GGC CGC GGC GGC GGC GGC GGG AAG CAG CGG GGC UGG GGU UCC AGG GGG AGC GGC CGC CGC CUC-3′ and for the G/A mutant 5′-GGC CGC GGC GGC GGC GGC G<u>A</u>G AAG CAG CG<u>A</u> AGC UG<u>A</u> <u>A</u>GU UCC AG<u>A</u> <u>A</u>AG AGC GGC CGC CGC CUC-3′. The sequences used were for wild-type MLL4 5′-GGC CCG CGG GUC CAG CGG GGC CGG GGA CGG GGU CGG GGC CGG GGC UGG GGC CCG AGU CGA GGC UG-3′ and for the G/A mutant 5′-GGC CCG CGG GUC CAG CG<u>A</u> <u>A</u>GC CG<u>A</u> <u>A</u>GA CG<u>A</u> <u>A</u>GU CG<u>A</u> <u>A</u>GC CG<u>A</u> <u>A</u>GC UG<u>A</u> <u>A</u>GC CCG AGU CGA GGC UG-3′.

## Potential G-quadruplex-forming sequences

The following RNA sequences were purchased from IDT.

**sc1** (G4 WT) 5′-GCUGC gg UGU gg AA gg AGU gg UC gg GUUGCGCAGCG-biotin-3′;

**sc1** (G4m) 5′-GCUGC aa UGU gg AA aa AGU gg UC gg GUUGCGCAGCG-biotin-3′;

**MLL1** (G4; 220–258) 5′-CGGC ggg AAGCAGC gggg CT gggg TTCCA gggg GAGCGG biotin-3′;

**MLL1** (G4m) 5′-CGGC aaa AAGCAGC gggg CT aaaa TTCCA gggg GAGCGG-biotin-3′;

**MLL4** (G4; 267–310) 5′-CCAGC gggg CC gggg AC gggg UC gggg CC gggg CU gggg CCCGA-biotin-3′;

**MLL4** (G4m) 5′-CCAGC <u>aaaa</u> CC gggg AC gggg UC <u>aaaa</u> CC gggg CU gggg CCCGA-biotin-3′.

## Immunofluorescence

U2OS cells grown on coverslips were washed twice with 1× PBS and fixed with 4% paraformaldehyde for 10 min at room temperature. The cells were then washed twice with 1× PBS and permeabilized with 0.5% Triton X-100 for 10 min. Permeabilization was followed by three washes with 1× PBS and cells were blocked with 10% FBS in PBS and labeled with primary antibody diluted in PBS containing 5% FBS. After three washes, the cells were labeled with Alexa Fluor 540 conjugated secondary antibody. DNA was counterstained with 4.6-diamidino-2-phenylindole (DAPI). The coverslips were washed three times with 1× PBS and mounted on slides using Immuno-Mount (Thermo Scientific, Waltham, MA). Images were captured using a Zeiss (Germany) M1 microscope with epifluorescence optics.

## Mass spectrometry and SILAC

Myc-Aven expressing HEK293T cells were lyzed and IP with anti-Myc epitope tagged antibodies. The IP proteins were resolved by SDS-PAGE, visualized by Coomassie Blue (SimplyBlue Safestain, Life Technologies), excised from the gel and digested with trypsin and subjected to LC/MS/MS analysis, as previously described (*Boisvert et al., 2012*). U2OS cells were cultured in DMEM (Dulbeccos's Modified Eagle Medium) depleted of arginine and lysine, as described previously (*Boisvert et al., 2012*). DMEM was supplemented with 10% of dialyzed FBS. Arginine and lysine were substituted with light (Arg0, Lys0), medium (Arg6, Lys4), or heavy (Arg10, Lys8) amino acids (Cambridge Isotope Laboratories, Inc., UK). The cells were cultured in the labeled medium for 6 passages for metabolic incorporation of the labeled amino acids. The light, medium, and heavy labeled cells were transfected with empty vector pcDNA3.1, FLAG-Aven, and FLAG-AvenΔRGG, respectively. Cell lysates were IP with anti-FLAG agarose beads and the IP complexes were mixed prior to mass spectrometry analysis.

Trypsin-digested peptides were separated using a Dionex Ultimate 3000 nanoHPLC system. 10 µl of sample (a total of 2 µg) in 1% (vol/vol) formic acid was loaded with a constant flow of 4 µl/min onto an Acclaim PepMap100 C18 column (0.3 mm id × 5 mm, Dionex Corporation, Sunnyvale, CA). After trap enrichment, peptides were eluted off onto a PepMap C18 nano column (75 µm × 50 cm, Dionex Corporation) with a linear gradient of 5–35% solvent B (90% acetonitrile with 0.1% formic acid) over 240 min with a constant flow of 200 nl/min. The HPLC (High-performance liquid chromatography) system was coupled to a QExactive mass spectrometer (Thermo Fisher Scientific Inc) via a EasySpray source. The spray voltage was set to 2.0 kV and the temperature of the column was set to 40°C. Full-scan MS survey spectra (m/z 350–1600) in profile mode were acquired in the Orbitrap with a resolution of 70,000 after accumulation of 1,000,000 ions. The ten most intense peptide ions from the preview scan in the Orbitrap were fragmented by collision-induced dissociation (normalized collision energy 35% and resolution of 17,500) after the accumulation of 50,000 ions. Maximal filling times were 250 ms for the full scans and 60 ms for the MS/MS scans. Precursor ion charge state screening was enabled and all unassigned charge states as well as singly, 7, and 8 charged species were rejected. The dynamic exclusion list was restricted to a maximum of 500 entries with a maximum retention period of 40 s and a relative mass window of 10 ppm. The lock mass option was enabled for survey scans to improve mass accuracy. Data were acquired using the Xcalibur software.

Data were processed, searched, and quantified using the MaxQuant software package version 1.4.1.2 as described previously (*Cox and Mann, 2008*) employing the Human Uniprot database (16/07/2013). The settings used for the MaxQuant analysis were 2 miscleavage were allowed; fixed modification was N-ethylmaleimide on cysteine; enzymes were Trypsin (K/R not before P); variable modifications included in the analysis were methionine oxidation and protein N-terminal acetylation. A mass tolerance of 7 ppm was used for precursor ions and a tolerance of 20 ppm was used for fragment ions. A maximum false positive rate of 1% was allowed for both peptide and protein identification.

## Methylation assays

GST-AvenRGG/RG was incubated with GST-tagged PRMT1 with 0.55 µCi of [methyl-$^3$H] S-adenosyl-L-methionine in the presence of 25 mM Tris-HCl (pH 7.5) for 1 hr at 37°C in a final volume of 25 µl. Reactions were stopped by adding 25 µl of 2× Laemmli buffer followed by boiling for 10 min. The proteins were separated by SDS-PAGE, stained with Coomassie Blue, and the destained gel was soaked in EN$^3$HANCE (Perkin Elmer Life Sciences, Waltham, MA), as per manufacturer's instructions and visualized by fluorography.

## Generating Aven⁻/⁻ cells using CRISPR/Cas9

HEK293T were co-transfected with a pEGFP-C1 (Clontech, Palo Alto, CA), a Cas9 expression vector (Addgene, Cambridge, MA) and expression plasmids encoding the following gRNAs; 5′-GGG GCC AGC GCG CCG GTA AGA GG-3′ and 5′-GCA GCG GCG GTA GCC AGA GGC GG-3′ targeting *Aven* exon 1. The gRNAs expression plasmids were synthesized by IDT (Coralville, IA), as described (*Mali et al., 2013*). Single cells expressing GFP were sorted using fluorescence-activated cell sorting several days after transfection and individual clones were expanded and screened by genomic PCR and by immunoblotting.

## Acknowledgements

We thank Janane Maheswaran and André Galarneau for their contributions on the initial observations to the Aven project. We acknowledge Koren Mann and Josée Hébert for critically reading the manuscript and helpful discussions. This work was supported by a grant from the Canadian Institute of Health Canada (MOP-67070) to SR and the Chaire de recherche de l'Université de Sherbrooke en strucucture de l'ARN et génomiquie to JPP. PT was funded, in part, by a James O and Maria Meadows studentship and with funds from the Fonds de recherche du Québec–Santé (FRQS) to SR. SGR holds an NSERC Fellowship.

## Additional information

### Funding

| Funder | Grant reference | Author |
|---|---|---|
| Canadian Institutes of Health Research (Instituts de recherche en santé du Canada) | MOP-67070 | Stéphane Richard |
| Fonds de Recherche du Québec - Santé (Fonds de la recherche en sante du Quebec) | | Stéphane Richard |

The funders had no role in study design, data collection and interpretation, or the decision to submit the work for publication.

### Author contributions

PT, JS, VG, F-MB, Conception and design, Acquisition of data, Analysis and interpretation of data, Drafting or revising the article; YC, Acquisition of data, Analysis and interpretation of data; SGR, J-MG, ZY, Acquisition of data, Analysis and interpretation of data, Drafting or revising the article; J-PP, IT, SR, Conception and design, Analysis and interpretation of data, Drafting or revising the article

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
