## [Decision Letter]

Thank you for sending your work entitled “Aven recognition of RNA G-quadruplexes regulates translation of the *Mixed Lineage Leukemia* protooncogenes” for consideration at *eLife*. Your article has been overall favorably evaluated by James Manley (Senior Editor), a Reviewing Editor, and two reviewers. One of the two reviewers, Ernesto Guccione, has agreed to share his identity.

The Reviewing Editor and the two reviewers, whose comments are below, discussed their reviews before reaching this decision, and the Reviewing Editor has assembled the following comments to help you prepare a revised submission.

The manuscript submitted by Thandapani et al. investigates the mechanism through which the protein Aven inhibits apoptosis leading to the promotion of malignancy. The authors present data supporting the role of Aven as an RNA binding protein through its N-terminal RGG/RG domain and suggest that it binds G-quadruplex RNA structures. Arginine methylation of the RGG/RG domain by PRMT1 was shown to affect RNA binding, association with polyribosomes, and interactions with two Tudor domain proteins (SMN and TDRD3). The authors identify G-quadruplex motifs in two mRNAs (MLL1 and MLL4) as targets of Aven. It also appears that the RGG/RG domain of Aven recruits an RNA helicase that can unwind G-quadruplexes (DHX36) to polyribosomes, and this increases the translation of mRNAs. Taken together the data suggest a model where Aven binds the G-quadruplex motifs in the MLL mRNAs, through its RGG/RG domain. It associates with polysomes dependent on methylation of its RGG/RG domain by PRMT1, and its binding to polysome-associated Tudor domain proteins. Polysome-associated Aven recruits a GQ-unwinding helicase that then promotes translation of MLL1 and MLL4 by destabilizing the G-quadruplexes in their coding sequences. Increased expression of these histone methyltransferases affects transcription of HOX mRNAs and this is associated with the increased cell proliferation and survival function postulated for Aven.

1) The most important issue with this manuscript has to do with incomplete or unclear quantitation of data, use of statistical tests that are not described at all, and lack of demonstration of reproducibility. For example, it needs to be indicated for every experiment how many times it has been performed independently. Blots should be quantified when possible. Error bars are almost never defined (an exception is Figure 5) and statistical tests are not described. There is no description of “n” numbers or how “n” is defined. It's difficult to evaluate the experiments presented without this information.

2) The paper is not very well referenced. There are missing references that should be cited and in some cases better papers could be cited. The most important of the missing references is Melzer et al. (Cell Death and Differentiation, 19:1435, 2012) in which they find cathepsin D cleavage sites and map the anti-apoptotic function of Aven to the C-terminus once released by proteolysis from the inhibitory N-terminus. I'd like the authors to address this work and discuss their findings in relation it.

3) The authors rely heavily on deletion of the ΔRGG region in Aven. It would be nice to have at least one experiment performed with an R-K mutant instead of the deletion construct.

Reviewer #1:

Figure 1 suggests that endogenous Aven can be methylated by endogenous PRMT1 on several arginines in the N-terminal RGG/RG domain, though a few definitive controls as mentioned below are missing. However, even without all the controls when all the data is considered together it's likely that the authors are correct.

However, relevant to panel (A), it's not correct to say that Aven has no other known domains (in the subsection “Aven RGG/RG motif is methylated by PRMT1”). The authors should cite, or defend why they aren't citing, several publications: one by Esmaili et al. (Cell Cycle, 9:3913, 2010) suggesting a CRM1-dependent NES, one by Hawley et al. (Open Biol. J., 5:6, 2012) finding a BH3 motif (binding to Bcl-X_L_), and one from Melzer et al. (Cell Death and Differentiation, 19:1435, 2012) who find cathepsin D cleavage sites and importantly map the anti-apoptotic function of Aven to the C-terminus once released by proteolysis from the inhibitory N-terminus? The authors should comment on these findings.

Figure 1 shows that transfected Myc-Aven can capture endogenous PRMT1 in HEK cells. The figure, legend and text (in the subsection “Aven RGG/RG motif is methylated by PRMT1”) all need modification to indicate that odd numbered lanes lack Myc-Aven.

Figure 1 uses OHT to knockout PRMT1 from a floxed allele in MEFs. The authors suggest that this shows that Myc-Aven is no longer methylated after loss of PRMT1 but a control is missing (mock transfection) which would show the methylated band in lane 1 is actually Myc-Aven and not a contaminating methylated PRMT1 substrate. This still would not control for a co-IPing methylated PRMT1 substrate dependent on the IP of Aven – transfection with Myc-Aven lacking the RGG/RG box would be definitive. Finally, MEFs lacking floxed PRMT1 would be an appropriate negative control to show that OHT alone doesn't inhibit methylation of Aven, i.e. show dependence on the PRMT1.

Figure 1: Endogenous Aven was IP'ed after tamoxifen-induced deletion of PRMT1, or not. Aven is shown to no longer be phosphorylated if PRMT1 is lost. Again it's possible that OHT treatment alone might lead to decreased Aven phosphorylation independent of PRMT1. Dependence on PRMT1 could be shown using MEFs lacking floxed PRMT1 as a negative control.

Figure 1—figure supplement 1: A line should be added to the text of the paper pointing out the conservation of methylated arginines across species.

Figure 2: the general conclusion is probably correct, that the Aven RGG/RG domain has the potential to bind G-quadruplex forming sequences in vitro even when it's unmethylated. Whether methylation changes the affinity to any degree can't be determined.

The RNA binding assays used in Figure 2 are not described in enough detail. For example, what was the volume of cell lysate to which 400-500 pmol of RNA was added (needed to calculate molarity of the RNA)?

Figure 2: The statement can't be made that methylated and unmethylated Aven bind equally well to sc1 from the experiment presented. Salt sensitivity is not the same as a Kd. At 150 mM salt either the protein or the RNA needs to be titrated and % bound at each concentration determined to calculate a Kd for methylated vs. unmethylated.

Figure 2: In this experiment a Kd could be calculated with the right assay since both peptides and RNA concentrations are known. The curves shown do not reach saturation so a solid conclusion about differing affinity can't be made. The method needs to be described including the sequences of the 4 RG motif peptides. RNA should be described as a concentration, not an amount.

Figure 2—figure supplement 1: Though quantitation and evidence of reproducibility are lacking, it does appear that deletion of the RGG box doesn't markedly affect phosphorylation of ATM or CHK2, or it's subcellular localization (though this is not a shuttling assay), or it's dimerization, though this is a double overexpression assay. A small note – the figure legend lists a different anti-pCHK2 antibody than in the figure labeling.

Figure 3 addresses whether Aven interacts with 3 Tudor domain proteins and whether this is dependent on the RGG box and its methylation status. In Figure 3, double overexpression GST pull-downs suggest the RGG box is required for TDRD3 and SMN. This is extended in (B) and (C) to the capture of endogenous TDRD3 and SMN and finally in (D) co-IP of endogenous Aven with endogenous TDRD3 and SMN is shown, and found to be dependent on OHT treatment. The authors suggest this means it's dependent on methylation of Aven by PRMT1 but dependence on PRMT1 is not shown. Further support for this would be OHT treated WT MEFS, or a second method of knocking down PRMT1, like siPRMT1 treatment. There is nothing in these studies to rule out complex formation via a third protein(s) or RNA that links Aven to the other two proteins, for example, if they are all polysome-associated, and intact polysomes are being IPed. RNAse treatment would be appropriate here. Note that sc1 is not a substitute for the RNAs that Aven might be interacting with in a cell.

Figure 4: The authors demonstrate that a percentage of cellular Aven is associated with polysomes that are actively translating based on the puromycin run-off experiment shown. However, the data that the RGG box affects polysome association is not convincing. If the blots in F and G were exposed a little more will they look like E? How many times was the experiment done? Why does rpS6 disappear completely with puromycin treatment? It should appear in the fractions with the monosome peak. Quantitation of the loss from polysomes with error bars would be appropriate to make the point here. In (K-M) the same concerns apply. As presented I don't find the decrease in polysome-associated Flag-Aven convincing.

Figure 5—figure supplement 1 should be referred to as table supplement 2 or vice versa. I am not clear about the reasoning behind or the significance of the statement that the mRNA encoding domains rich in R's and G's may have propensity to form G-quadruplexes. Nor why that particular consensus was used to query ORFs for potential G-quadruplex forming sequences. Is there literature to support that that query predicts that RNA fold? The rationale should be made clear. I'm particularly worried about the inclusion of C's without restraints in the N1-7 intervals. A stretch of C's will almost certainly form a duplex with nearby runs of G's interrupting the potential for GQ formation. A folding program could supplement this analysis, as could conservation across species that would predict maintenance of the GQ fold. Compensatory mutation to preserve folding would be even stronger evidence. If it's correct that the C-terminus of Aven holds the anti-apoptotic function once released from the N-terminus, it's not clear why the authors are proposing the connection between PG4 binding in mRNAs encoding proteins involved in leukemic cell survival. Could the authors please comment on the findings in Melzer et al. (Cell Death and Differentiation, 19:1435, 2012)?

Figure 5: Again, this is not a proper RNA binding assay (Figure 5). The finding that Aven-ΔRGG no longer binds RNA is not evidence for a specific interaction (Figure 5). The positive charge of the RGG box will lead to nonspecific RNA association under certain conditions, again it depends on the Kd and that data is not available.

Figure 5: In the CLIP experiments presented in this panel, there is no evidence for a direct interaction. A non-crosslinked control is missing. In addition, the “CLIP” technique as described was done under conditions that the same reference (Huppertz et al.) concluded caused substantial cell toxicity, and no washes of the IP are described. It should not be referred to as “CLIP” in the current form, since there is no attempt to take advantage of the covalent bond introduced by crosslinking which would include rigorous purification of Aven:RNA complexes including stringent washes and SDS-PAGE. Moreover, this is PAR-CLIP, not iCLIP as referenced. The reference should be changed to the methodology developed by the Tuschl lab. An unbiased map of where Aven is bound to RNA in vivo could be obtained by performing the whole CLIP protocol including high throughput sequencing of the RNA tags. However, overexpression of Aven with a Flag tag is not optimal, as RBPs tend to associate with targets of lesser affinity and specificity when overexpressed beyond normal stoichiometry with their RNA targets. The antibody against Aven could be used to perform CLIP on endogenous Aven with unbiased and more comprehensive results.

Figure 7: Panel (B) shows polysome profiles for WT and Aven-null cultured cells. The conclusion is made that the Aven-null cells have less polysomes, but it appears that they have less total ribosomes as well since the monosome peak is decreased. If equal cell numbers were used for the polysome analysis this might suggest a decrease in the total machinery available for protein synthesis. Panels (C)-(F) show the results for only part of the gradient. It would be valuable to see the whole gradient. Moreover, mRNA per fraction can only be plotted as a percentage of the total RNA if the total is determined for the whole gradient. For example, in (C) a large increase in MLL1 mRNA in the WT (blue line) in fractions 1-4 could cause the blue line in fractions 13-14 to actually be less than the red line, which changes the conclusions. If the authors mean that they quantified the mRNA in an aliquot of the lysate applied to the gradient, this should be clarified. For less confusion, real values for each fraction should be shown rather than percentage, which makes the assumption that the total mRNA levels in the presence or absence of Aven are unchanged. It would be nice to see this data again as it supports Figure 6.

Figure 8: Again, please include reproducibility (n) and statistical tests and whether the results are statistically robust.

Figure 9: Could the authors comment on whether the small global decrease in initiation in the presence of Flag-Aven is reproducible? (Blue line, loss of polysomes with a gain in the subunits and monosomes not seen for the green and red lines). In (D), where does the DHX36 go if not on polysomes? There doesn't seem to be any difference in total levels in panel (A). Does the blot need to be exposed longer? Fractions 1-4, maybe; it would be nice to see this. It does look like DHX36 has normal polysome association in (E) but I'm again unconvinced that there is a significant difference between the results in panels (E) and (F). Quantitation? Reproducible?

Reviewer #2:

1) Add molecular markers also on the right panel in Figure 1.

2) could be moved to supplementary figures.

3) One of the key figures in my opinion is Figure 7, where the authors are able to show that the MLL1 and MLL4 are selectively depleted from the heavy polysome fractions. This is performed in the presence or absence of Aven. It would be great to see the same experiment done in siPRMT1 and/or siSMN+siTDR3 conditions. An alternative suggestion would be to move this figure at the end of the manuscript and add also siDHX36.

4) The data presented in Figure 10 do not add much and could be removed (or improved). Questions that come to mind are: is the drug selectively killing cancer cells that overexpress MLL1/MLL4? Or does it kill any cell? What is the toxicity of this drug? How direct is the effect we see on MLL1/MLL4 protein abundance, as opposed to a general toxicity in cell culture?

[Editors’ note: this article was subsequently rejected after discussions between the reviewers, but the authors were invited to resubmit after an appeal against the decision.]

Thank you for sending the revised version of your work entitled “Aven recognition of RNA G-quadruplexes regulates translation of the *Mixed Lineage Leukemia* protooncogenes” for consideration at *eLife*. Your revised submission has been evaluated by James Manley (Senior Editor), a Reviewing Editor, and one peer reviewer, and the decision was reached after discussions between the Reviewing Editor and the reviewer. Based on our discussions and the individual review below, we regret to inform you that your work cannot be considered further for publication in *eLife*.

While the editor and the reviewer felt that you had improved many aspects of the manuscript that had been identified as problematic (some new data and clarification of the number of independent replicates and explanation of statistical tests), they also felt that your work still does not sufficiently connect your observations to the biology of the system. The editor and the referee felt that, while many of the details of the function of Aven were defined, the key question as to the biological significance of the RGG box (and its potential interaction with RNA and the downstream consequences) was not adequately addressed. In light of these deficiencies, it was agreed that your manuscript fails to meet the level of significance required for publication in *eLife*.

Reviewer #1:

In this resubmission the authors have improved the paper considerably with new data and clarification of the number of independent replicates and explanation of statistical tests. Overall, they have addressed most of my experimental concerns.

However, there are two bigger issues that are still bothering me. The background to the paper is that the authors previously found that Aven harbors an N-terminal RGG motif conserved across species. It strikes me that the most important issue, before investigating its function, should be to show that the RGG box is required for the anti-apoptotic (or other) biological functions of Aven in affecting cancer progression. Once established, it then becomes of greater interest to understand the molecular functions of the RGG box. My reading of the paper is that those assays haven't been done and in terms of the ones that have been performed, deletion of the RGG box had no effect. Moreover, as I pointed out in my first review, Melzer et al. have published that the anti-apoptotic function of Aven resides in the C-terminus and the N-terminus (containing the RGG box) acts to repress this activity until cleavage of Aven by cathepsin D releases the active C-terminus. I think this at odds with the idea that the RGG box is central to the anti-apoptotic activity unless I'm missing something. I asked the authors to discuss this and they didn't answer the question. They do add a sentence or two to the paper (in addition to agreeing to mention the Melzer et al. work) in which they say they can barely detect the product of cathepsin D cleavage, therefore it's unlikely to be important to what they are studying, but that doesn't address the issue of demonstrating that the N-terminal RGG box is actually required for the anti-apoptotic properties of Aven.

I think they have skipped the step, or are avoiding the issue, of establishing that the RGG box is important. Because they bioinformatically found an RGG box in Aven, and some RGG boxes bind RNA, and some are methylated by PRMT1, and some have been reported to bind GQ's in vitro, and some GQ's have been identified in RNA, and in some cases may affect translation, the authors have built a story suggesting that this is true for Aven and is relevant for understanding its anti-apoptotic function. They do have good data for a lot of this now, except for the importance of the RGG box in the functions of Aven that make it interesting.

If they had established that the RGG box was critical, this would still be a higher impact paper if they had then used PAR-CLIP or a similar assay to 1) determine if Aven binds RNA in vivo, and 2) generate an unbiased genome wide map of where it binds. The binding sites would not only reveal mRNA targets but would show GQ motif consensus if that's really the in vivo binding site for Aven. The polysome-association and PRMT1 methylation studies then follow to determine the outcome of Aven binding those sites in those mRNAs (which may or may not be translational control) as well as what regulates it. Instead, the paper reads like a very biased expedition to explore a few reported properties of RGG boxes and RNA G-quadruplexes based on prior literature, not all of which is high quality and many conclusions are from in vitro studies with unclear in vivo relevance. I realize we previously discussed this and agreed that it would be unreasonable to ask them to do these experiments (though it does bother me that they have already done half the PAR-CLIP experiment but didn't finish it, which could transform their paper (or prove their assertions correct) fairly easily).

The work is appealing because the authors present a compelling argument that understanding how Aven inhibits apoptosis in cancer cells, and consequently allows their progression when it's overexpressed, has potential for increasing therapeutic avenues for some human cancers. But I'm worried about their reticence to discuss the Melzer paper in regard to the issue of the RGG box being required for the anti-apoptotic function. Given the weak experimental approach (which we agreed not to ask them to change) I don't think it’s unreasonable to at least ask them to fully address evidence in support or in conflict with the idea that the RGG box has a relevant biologic role, to validate their experiments that speak to its function.

[Editors’ note: what now follows is the decision letter after the authors submitted for further consideration.]

Thank you for submitting your work entitled “Aven recognition of RNA G-quadruplexes regulates translation of the *Mixed Lineage Leukemia* protooncogenes” for further consideration at *eLife*. Your article and letter of appeal have been evaluated by James Manley (Senior Editor), a Reviewing Editor, and two reviewers. One of the two reviewers (Reviewer 1) reviewed the earlier versions of your work, while the other (Reviewer 3) has not reviewed this study before and provides a fresh perspective.

Although the ultimate biological relevance of the RGG box, or even translation regulation by Aven, is not yet clear, we are willing to move forwards with acceptance, providing you can address the minor comments of Reviewer 3. We also ask you to provide a final response to Reviewer 1's comments and address them where you can within the Discussion. We are not asking for additional experiments.

Reviewer #1 (abridged):

In their appeal Dr. Richard and co-authors discuss two of the points I raised when re-reviewing the manuscript.

The most important issue is with the second point I raised. To paraphrase my previous summary of that issue, in order to support the importance of this work investigating the function of the Aven RGG box, it should be demonstrated that the RGG box is required for the anti-apoptotic (or other) biological functions of Aven in affecting cancer progression. Once that's established, it then becomes of greater interest to understand the molecular functions of the RGG box. In general those assays haven't been done and in terms of the ones that have been performed, deletion of the RGG box had no effect. Moreover, as I pointed out in my first review, Melzer et al. have published that the anti-apoptotic function of Aven resides in the C-terminus and the N-terminus (containing the RGG box) acts to repress this activity until cleavage of Aven by cathepsin D releases the active C-terminus. This is at odds with the idea that the RGG box is central to the anti-apoptotic activity.

In their appeal, they have provided a greater response to that but are still missing/avoiding the point and lacking in the validation experiments that could support the importance of the function of the RGG box.

The authors cite literature that many groups agree that full-length Aven has anti-apoptotic activity. However, my concern isn't so much whether full length Aven has an anti-apoptotic role but the other finding of Melzer et al.: that robust anti-apoptotic activity can be mapped to fragment of Aven lacking the RGG box, which would be at odds with the current paper finding that the RGG box is mediating the anti-apoptotic activity. What I wanted the Richard group to discuss is the Melzer et al. finding that the RGG-box containing N-terminus of Aven mediates homodimerization, and that its release from FL Aven by cathepson D cleavage activates the C-terminus as an anti-apoptotic molecule. Whether or not the N-terminus is really inhibitory, or FL Aven has anti-apoptotic activity, the Richard group should provide some insight as to why Melzer et al. find that the RGG deleted form of Aven has anti-apoptotic activity.

The authors now include a new figure for the reviewer (Author response image 1) using an assay for Aven function which is clonogenic survival of U2OS cells, purporting to show that the RGG box is required. The assay shows that without endogenous Aven, the cells are killed with increasing UV irradiation. Overexpression of Aven protects them. There is a small decrease in the ability of Aven to do so if the RGG box is deleted but it is only significant at the highest UV dose and is a modest effect, perhaps 2-fold. Panel B shows a Western blot of the expression of Aven and Aven-ΔRGG and it is quite clear that there's much more WT Aven produced than Aven-ΔRGG. It is impossible to know if the effects they see are due to Aven dose or to the RGG box. CRISPR-mediated deletion of the RGG box from endogenous Aven would be more compelling. They've used that technology to make Aven null cells and perhaps could use the same technique to make an elegant RGG null form of Aven.

The authors also cite three figures in their paper (Figures 3, 7 and 10) that they claim support the role of the RGG box, however, Figure 3 only shows the RGG box is needed for protein-protein association. Figure 10 shows that the ability of Aven to regulate translation of a luciferase reporter containing a G-quadruplex forming sequence is dependent on the RGG box. Nicely shown, but very in vitro and not an assay of cell survival. Figure 7 is quoted by the authors as relevant, but I think they mean Figure 5 G-H which are the RIP and partly finished PAR-CLIP experiment to show the RGG box is important for association with MLL1 and MLL4 RNA.

As I wrote previously, I think the Richard group has “skipped the step, or are avoiding the issue, of establishing that the RGG box is important […]. They do have good data for a lot of this now, except for the importance of the RGG box in the functions of Aven that make it interesting.”

Reviewer #3:

This article concerns the role of Aven in regulating the translation of leukemia-relevant mRNAs. Aven is an RGG box protein that promotes cell survival and is linked to osteosarcoma and AML. The strength of the manuscript is the use of multiple lines of evidence to justify the majority of the conclusions. The weakness is the qualitative nature of many of the assays. Nevertheless, all of the data presented are internally consistent, and I am left with the impression that the data as presented do largely support the authors' conclusions. Both mechanistic and biological insights arise from the study, which increases overall interest to a broad readership. While the manuscript would be improved by a more thorough study of Aven-associated mRNAs through a true PAR-CLIP study, or by further studies into the biological relevance of the handful of newly identified regulated mRNAs presented here, I agree with the authors' rebuttal that addition of such data to this manuscript would increase the scope well beyond what should be found in a single paper.

Ultimately, it is interesting that Aven binds to G-quartets, that its association with polysomes is methylation dependent, and that it regulates the translation of at least a few leukemia specific mRNAs, ultimately leading to reduced amounts of *HOX* gene transcription. And while a broader study would enhance the impact, it is hard to imagine filling this manuscript full of even more experiments.

Minor comments:

1) It seems disingenuous to call the photocrosslinking IP experiments presented in this paper “PAR-CLIP”. The use of photoactivatable nucleotides to crosslink proteins to RNA to assist in recovery by IP preceded the paper by Hafner et al. by many years. PAR-CLIP, as published, is touted as a transcriptome-wide approach.

2) The description of the RGG box as an “RNA Recognition Motif” (in the Introduction) could be misleading, as the RRM is a specific RNA-binding domain that is not the same as the RGG box. Though many proteins contain both.

3) The rationale behind the selection of SMN and TDRD3 for further study was not clear to this reviewer. Have either been implicated in osteosarcoma or myeloid leukemia? There are 71 Tudor domain proteins in humans according to EMBL SMART database.

4) Can sc1 RNA bind to Aven if it hasn't been induced to fold into a G-quartet?

5) What fraction of the proteins associated with Aven in the SILAC experiments are RNA-dependent?

6) I am surprised that real-time PCR was used to detect enrichment of the G-quadruplex structures in the photocrosslinked IPs. I would have guessed that the G quadruplex would have prevented efficient reverse transcription. The authors should comment on this point.

7) The significant digits in Figure 2 should be re-evaluated.

---

## [Author Response]

[Editors’ note: the author responses to the first round of peer review follow.]

*1) The most important issue with this manuscript has to do with incomplete or unclear quantitation of data, use of statistical tests that are not described at all, and lack of demonstration of reproducibility. For example, it needs to be indicated for every experiment how many times it has been performed independently. Blots should be quantified when possible. Error bars are almost never defined (an exception is*
Figure 5*) and statistical tests are not described. There is no description of “n” numbers or how “n” is defined. It's difficult to evaluate the experiments presented without this information*.

We now provide additional quantification and statistical tests, and indicate the number of independent experiments performed.

*2) The paper is not very well referenced. There are missing references that should be cited and in some cases better papers could be cited. The most important of the missing references is Melzer et al. (Cell Death and Differentiation 19:1435 (2012)) in which they find cathepsin D cleavage sites and map the anti-apoptotic function of Aven to the C-terminus once released by proteolysis from the inhibitory N-terminus. I'd like the authors to address this work and discuss their findings in relation it*.

We have added additional functional regions in the diagram of Aven protein in Figure 1 including a putative BH3 domain identified using bioinformatics and a functional nuclear export signal (NES). We also cite additional pertinent papers, such as [44]; [61] and [36].

[61] describe that in addition to the full length Aven (MW∼50kDa), MCF-7 and HEK293T cells express a truncated Aven C-terminal protein (MW∼30kDa) that is thought to be generated by Cathepsin D cleavage. It also appears that this Aven C-terminal fragment has potent anti-apoptotic activity, as judged by overexpression studies. We used the same ProScience antibody as [61] which recognizes the Aven C-terminus. An ∼30kDa Aven C-terminal protein was notably absent in MOLT-4 (Figure 6) and CCRF-CEM (Figure 6) cells. In HEK293T cells, however, after prolonged exposure, we do see a very minor band (∼34kDa) that is absent in the Aven CRISPR/CAS null cells that could represent cleaved C-terminal Aven fragment described by Melzer et al.

To reflect these findings, we have added the following text in the Results section: “A minor Aven ∼34kDa fragment (Figure 7, denoted by asterisk) was observed in HEK293T cells and likely represents a cathepsin D cleaved fragment reported previously [ #8127]”. We have also added the following to the Discussion: “It was reported that an N-terminal deleted fragment of Aven cleaved by cathepsin D harbors […]. Thus cathepsin D mediated cleavage of Aven is unlikely involved in the regulation of translational control described herein.”

*3) The authors rely heavily on deletion of the ΔRGG region in Aven. It would be nice to have at least one experiment performed with an R-K mutant instead of the deletion construct*.

We have generated Aven R-K mutant and present data in Figure 10 which shows that this mutant closely mimics effects observed in experiments using Aven RGG.

Reviewer #1:

Figure 1
*suggests that endogenous Aven can be methylated by endogenous PRMT1 on several arginines in the N-terminal RGG/RG domain, though a few definitive controls as mentioned below are missing. However, even without all the controls when all the data is considered together it's likely that the authors are correct*.

*However, relevant to panel (A), it's not correct to say that Aven has no other known domains (in the subsection “Aven RGG/RG motif is methylated by PRMT1”). The authors should cite, or defend why they aren't citing, several publications: one by Esmaili et al. (Cell Cycle, 9:3913, 2010) suggesting a CRM1-dependent NES, one by Hawley et al. (Open Biol. J., 5:6, 2012) finding a BH3 motif (binding to Bcl-X*_*L*_*), and one from Melzer et al. (Cell Death and Differentiation, 19:1435, 2012) who find cathepsin D cleavage sites and importantly map the anti-apoptotic function of Aven to the C-terminus once released by proteolysis from the inhibitory N-terminus? The authors should comment on these findings.*

Please see response to 1) above.

Figure 1
*shows that transfected Myc-Aven can capture endogenous PRMT1 in HEK cells. The figure, legend and text (subsection “Aven RGG/RG motif is methylated by PRMT1”, second paragraph) all need modification to indicate that odd numbered lanes lack Myc-Aven*.

The legend for Figure 1 and the Results section were edited, as suggested.

The Results section was also modified to include the following: “PRMT1 was present in anti-Myc immunoprecipitates of the Myc-Aven transfected cells, but not in the empty vector transfected cells (Figure 1, lanes 3, 4).”

Figure 1
*uses OHT to knockout PRMT1 from a floxed allele in MEFs. The authors suggest that this shows that Myc-Aven is no longer methylated after loss of PRMT1 but a control is missing (mock transfection) which would show the methylated band in lane 1 is actually Myc-Aven and not a contaminating methylated PRMT1 substrate. This still would not control for a co-IPing methylated PRMT1 substrate dependent on the IP of Aven – transfection with Myc-Aven lacking the RGG/RG box would be definitive. Finally, MEFs lacking floxed PRMT1 would be an appropriate negative control to show that OHT alone doesn't inhibit methylation of Aven, i.e. show dependence on the PRMT1*.

Myc-Aven migrates ∼70kDa, as it was engineered to harbor 5 Myc-tags to distinguish it from endogenous Aven that normally migrates as ∼50kDa. Thus, we are confident of the identity of the Myc-Aven band, however, we have added an anti-Myc immunoblot (Figure 1; panel below ASYM25) of the immunoprecipitated proteins to demonstrate that the band at ∼70kDa is indeed Myc-Aven.

Figure 1*: Endogenous Aven was IP'ed after tamoxifen induced deletion of PRMT1, or not. Aven is shown to no longer be phosphorylated if PRMT1 is lost. Again it's possible that OHT treatment alone might lead to decreased Aven phosphorylation independent of PRMT1. Dependence on PRMT1 could be shown using MEFs lacking floxed PRMT1 as a negative control.*

To address the question related to potential inadvertent effects caused by the use of OHT, we now add an alternative approach to show hypomethylation of Aven. HEK293T cells were transfected with Myc-Aven and depleted of PRMT1 using siRNAs (new Figure 1). We obtain similar data as using floxed PRMT1 MEFs treated with OHT i.e. that Myc-Aven is recognized by ASYM25b in PRMT1-proficient, but not PRMT1-deficient cells. These data demonstrate that Aven is indeed a substrate of PRMT1 in vivo.

Figure 1—figure supplement 1*: A line should be added to the text of the paper pointing out the conservation of methylated arginines across species*.

In Figure 1—figure supplement 1 we have boxed the conserved arginines. The following sentences were added to the Results section: “Arginines R5, R8, R11, R37, R50 and R63 are also conserved in murines (Figure 1—figure supplement 1). Taken together, these findings demonstrate that the Aven RGG/RG motif is methylated by PRMT1 on conserved arginines.”

Figure 2*: The general conclusion is probably correct, that the Aven RGG/RG domain has the potential to bind G-quadruplex forming sequences* in vitro *even when it's unmethylated. Whether methylation changes the affinity to any degree can't be determined*.

*The RNA binding assays used in*
Figure 2
*are not described in enough detail. For example, what was the volume of cell lysate to which 400-500 pmol of RNA was added (needed to calculate molarity of the RNA)?*

Figure 2*: The statement can't be made that methylated and unmethylated Aven bind equally well to sc1 from the experiment presented. Salt sensitivity is not the same as a Kd. At 150 mM salt either the protein or the RNA needs to be titrated and % bound at each concentration determined to calculate a Kd for methylated vs. unmethylated*.

Figure 2*: In this experiment a Kd could be calculated with the right assay since both peptides and RNA concentrations are known. The curves shown do not reach saturation so a solid conclusion about differing affinity can't be made. The method needs to be described including the sequences of the 4 RG motif peptides. RNA should be described as a concentration, not an amount*.

To address these issues, we performed a new set of experiments where we obtain a Kd for two separate Aven RGG/RG peptides (see Figure 2 and Figure 2—figure supplement 1 panel D). Arginine methylation did not affect the binding affinities, as the Aven TriRG peptides (Kd ∼80-90nM; Figure 2) and the DiRGG peptides (Kd∼173-200nM; Figure 2—figure supplement 1 panel D) did not have different Kds.

Figure 2—figure supplement 1*: Though quantitation and evidence of reproducibility are lacking, it does appear that deletion of the RGG box doesn't markedly affect phosphorylation of ATM or CHK2, or it's subcellular localization (though this is not a shuttling assay), or it's dimerization, though this is a double overexpression assay. Small note – the figure legend lists a different anti-pCHK2 antibody than in the figure labeling*.

The activation of ATM and CHK2 and the cytoplasmic localization was performed n = 3. We have clearly stated the number of independent replicates that were performed throughout the revised version of the manuscript, including the legend of Figure 2—figure supplement 1. We have corrected the pCHK2 antibody to be pT68.

Figure 3
*addresses whether Aven interacts with 3 Tudor domain proteins and whether this is dependent on the RGG box and its methylation status. In*
Figure 3*, double overexpression GST pull-downs suggest the RGG box is required for TDRD3 and SMN. This is extended in (B) and (C) to the capture of endogenous TDRD3 and SMN and finally in (D) co-IP of endogenous Aven with endogenous TDRD3 and SMN is shown, and found to be dependent on OHT treatment. The authors suggest this means it's dependent on methylation of Aven by PRMT1 but dependence on PRMT1 is not shown. Further support for this would be OHT treated WT MEFS, or a second method of knocking down PRMT1, like siPRMT1 treatment. There is nothing in these studies to rule out complex formation via a third protein(s) or RNA that links Aven to the other two proteins, for example, if they are all polysome-associated, and intact polysomes are being IPed. RNAse treatment would be appropriate here. Note that sc1 is not a substitute for the RNAs that Aven might be interacting with in a cell*.

We agree that there may be a third protein or RNA in the complex. To clarify these issues we added the following sentence to the Discussion: “The Aven interaction with TDRD3 and SMN may require other protein or RNA components in the complex for enhanced association.”

As suggested by the reviewer, we have performed a new experiment (Figure 3), showing that siPRMT1 disrupts the interaction between FLAG-Aven and endogenous SMN and TDRD3.

Figure 4*: The authors demonstrate that a percentage of cellular Aven is associated with polysomes that are actively translating based on the puromycin run-off experiment shown. However, the data that the RGG box affects polysome association is not convincing. If the blots in F and G were exposed a little more will they look like E? How many times was the experiment done? Why does rpS6 disappear completely with puromycin treatment? It should appear in the fractions with the monosome peak. Quantitation of the loss from polysomes with error bars would be appropriate to make the point here. In (K-M) the same concerns apply. As presented I don't find the decrease in polysome-associated Flag-Aven convincing*.

To address the reviewer’s concerns, we now present both short and long exposures of the immunoblots for FLAG-Aven and FLAG-Aven RGG (Figure 4) and have modified the text accordingly to addresses exposure time and reproducibility. We also show quantification of these data by densitometry (Figure 4–figure supplement 2). We have also edited the legend for Figure 4.

Figure 5—figure supplement 1
*should be referred to as table supplement 2 or vice versa. I am not clear about the reasoning behind or the significance of the statement that the mRNA encoding domains rich in R's and G's may have propensity to form G-quadruplexes. Nor why that particular consensus was used to query ORFs for potential G-quadruplex forming sequences. Is there literature to support that that query predicts that RNA fold? The rationale should be made clear. I'm particularly worried about the inclusion of C's without restraints in the N1-7 intervals. A stretch of C's will almost certainly form a duplex with nearby runs of G's interrupting the potential for GQ formation. A folding program could supplement this analysis, as could conservation across species that would predict maintenance of the GQ fold. Compensatory mutation to preserve folding would be even stronger evidence. If it's correct that the C-terminus of Aven holds the anti-apoptotic function once released from the N-terminus, it's not clear why the authors are proposing the connection between PG4 binding in mRNAs encoding proteins involved in leukemic cell survival. Could the authors please comment on the findings in Melzer et al*. *(Cell Death and Differentiation, 19:1435, 2012)?*

We now refer to the table as Figure 5—figure supplement 1. We have edited our statements to clarify why we searched for PG4: “Next, we reasoned that since RGG/RG motifs are encoded by G-rich sequences […] [fx1]and it revealed ∼1600 PG4 in human ORFs (Figure 5—figure supplement 1).”

We calculated cG/cC score and the RNAfold using v2.1.7 and these data are added in (Figure 5—figure supplement 1). To reflect this, the following text was added to the Results section: “We also provide a cG/cC score […] predicting G4 formation than the cG/cC ratio [Beaudoin, 2014 #8043].”

Figure 5*: Again, this is not a proper RNA binding assay (*Figure 5*). The finding that Aven-ΔRGG no longer binds RNA is not evidence for a specific interaction (*Figure 5*). The positive charge of the RGG box will lead to nonspecific RNA association under certain conditions, again it depends on the Kd and that data is not available*.

We have modified the text accordingly (“We examined whether Aven associated with the PG4 RNA […] in Figure 2 with a fluoresceinated PG4 (sc1) to be in the 80-200nM range).

Figure 5*: In the CLIP experiments presented in this panel, there is no evidence for a direct interaction. A non-crosslinked control is missing. In addition, the “CLIP” technique as described was done under conditions that the same reference (Huppertz et al.) concluded caused substantial cell toxicity, and no washes of the IP are described. It should not be referred to as “CLIP” in the current form, since there is no attempt to take advantage of the covalent bond introduced by crosslinking which would include rigorous purification of Aven:RNA complexes including stringent washes and SDS-PAGE. Moreover, this is PAR-CLIP, not iCLIP as referenced. The reference should be changed to the methodology developed by the Tuschl lab. An unbiased map of where Aven is bound to RNA* in vivo *could be obtained by performing the whole CLIP protocol including high throughput sequencing of the RNA tags. However, overexpression of Aven with a Flag tag is not optimal, as RBPs tend to associate with targets of lesser affinity and specificity when overexpressed beyond normal stoichiometry with their RNA targets. The antibody against Aven could be used to perform CLIP on endogenous Aven with unbiased and more comprehensive results*.

We have edited the Methods section to indicate that we performed PAR-CLIP, not iCLIP. Indeed, our cells were healthy after the 8 h 4SU treatment. We have also edited the reference (Hafner et al., 2010 rather than ; [45]) and the total of 4 washes were performed, which is stated in the revised Methods section.

Moreover, as suggested we have performed an additional PAR-CLIP experiment with or without UV crosslinking and the IPs were performed with rabbit control IgG or anti-Aven (Proscience) antibodies. We observe binding of Aven to PG4 sequences and only when UV is utilized (Figure 5).

Figure 7*: panel (B) shows polysome profiles for WT and Aven-null cultured cells. The conclusion is made that the Aven-null cells have less polysomes, but it appears that they have less total ribosomes as well since the monosome peak is decreased. If equal cell numbers were used for the polysome analysis this might suggest a decrease in the total machinery available for protein synthesis. Panels (C)-(F) show the results for only part of the gradient. It would be valuable to see the whole gradient. Moreover, mRNA per fraction can only be plotted as a percentage of the total RNA if the total is determined for the whole gradient. For example, in (C) a large increase in MLL1 mRNA in the WT (blue line) in fractions 1-4 could cause the blue line in fractions 13-14 to actually be less than the red line, which changes the conclusions. If the authors mean that they quantified the mRNA in an aliquot of the lysate applied to the gradient, this should be clarified. For less confusion, real values for each fraction should be shown rather than percentage, which makes the assumption that the total mRNA levels in the presence or absence of Aven are unchanged. It would be nice to see this data again as it supports*
Figure 6.

We used an identical number of cells and loaded the same amount of material (i.e. 10 ODs at 254nm) on the gradients. We then integrated the area under curves and normalized curves based on the observed differences in the area, which is a standard method of normalization of polysome absorbance profiles (Gandin et al., JOVE, 2014). Aven null cells exhibited comparable absorbance profiles as Aven-proficient cells (Figure 7). To accommodate the reviewer’s request, the absorbance profiles for the entire gradient are now presented throughout the manuscript. Also, we observe a clear shift of the MLL1 and MLL4 mRNAs, but not β-actin into the lighter polysomal fractions in cells lacking Aven as compared to Aven-proficient cells (Figure 7). We also include new data showing that siPRMT1 and siDHX36 induces similar shifts in MLL1 and MLL4 mRNAs towards lighter polysomal fractions, but exert only a marginal effect on polysomal loading of β-actin mRNA, as compared to control cells (Figure 8; Figure 9). To analyze qPCR data we used relative standard curve method as described in ABI User Bulletin # 4.

Figure 8*: Again, please include reproducibility (n) and statistical tests and whether the results are statistically robust*.

The following information is now provided in the legend of Figure 10 (was Figure 8 previously), panels B and C.

“Error bars represent standard deviation values. The experiments were performed three independent times (n = 3) and each independent experiment was performed in technical triplicates. The significance was measured by ANOVA followed by post hoc comparison using Tukey test. **p* < 0.05.” (Legend of Figure 10)

“Error bars represent standard deviation values. The experiments were performed three independent times (n = 3) and each independent experiment was performed in technical triplicates. The significance was measured by ANOVA followed by post hoc comparison using Tukey test. **p* < 0.05, n.s. non-significant.” (Legend of Figure 10)

Figure 9*: Could the authors comment on whether the small global decrease in initiation in the presence of Flag-Aven is reproducible? (Blue line, loss of polysomes with a gain in the subunits and monosomes not seen for the green and red lines). In (D), where does the DHX36 go if not on polysomes? There doesn't seem to be any difference in total levels in panel (A). Does the blot need to be exposed longer? Fractions 1-4, maybe; it would be nice to see this. It does look like DHX36 has normal polysome association in (E) but I'm again unconvinced that there is a significant difference between the results in panels (E) and (F)*. *Quantitation? Reproducible?*

Combining the comments from both reviewers, we have performed additional polysomal fractionations and we monitored the levels of MLL1 and MLL4 mRNAs in each fraction. We observe a striking difference in cells depleted of DHX36 (new Figure 9). This combined with the luciferase assay builds a strong case that DHX36 stimulates translation of MLL1 and MLL4 mRNAs.

Reviewer #2:

*1) Add molecular markers also on the right panel in*
Figure 1.

Molecular mass markers were added to the right panel as well.

*2)*
Figure 4
*Panel C and D could be moved to supplementary figures*.

Panels C and D were moved to Figure 4—figure supplement 1.

*3) One of the key figures in my opinion is*
Figure 7*, where the authors are able to show that the MLL1 and MLL4 are selectively depleted from the heavy polysome fractions. This is performed in the presence or absence of Aven. It would be great to see the same experiment done in siPRMT1 and/or siSMN+siTDR3 conditions. An alternative suggestion would be to move this figure at the end of the manuscript and add also siDHX36*.

As suggested by the reviewer, we have reorganized our figures and we now show that MLL1 and MLL4 are selectively depleted from heavy polysomes fractions with siPRMT1 (new Figure 8) and siDHX36 (new Figure 9). We also repeated the polysomal fractionation with siSMN and siTDRD3 (Figure 4).

*4) The data presented in*
Figure 10
*do not add much and could be removed (or improved). Questions that come to mind are: is the drug selectively killing cancer cells that overexpress MLL1/MLL4? Or does it kill any cell? What is the toxicity of this drug? How direct is the effect we see on MLL1/MLL4 protein abundance, as opposed to a general toxicity in cell culture?*

Figure 10 was removed, as we agree with the reviewer that the toxicity of the G4 ligand drug may be due to the presence of many other G4 motifs present in the genome.

[Editors' note: the author responses to the second round of peer review follow.]

We would like to appeal to the rejection of our manuscript entitled “Aven recognition of RNA Gquadruplexes regulates translation of the *Mixed Lineage Leukemia* protooncogenes ”. Reviewer #2 found our manuscript suitable for presentation, whereas Reviewer #1 found that we were largely responsive to her/his comments, and that the revised version of the manuscript was vastly improved. Reviewer#1 felt, however, that our revised manuscript was not suitable for publication in *eLife*, which she/he rationalized by two major points:

*I) “I realize we previously discussed this and agreed that it would be unreasonable to ask them to do these experiments (though it does bother me that they have already done half the PAR-CLIP experiment but didn't finish it, which could transform their paper (or prove their assertions correct) fairly easily). […] Given the weak experimental approach (which we agreed not to ask them to change) I don't think it’s unreasonable to at least ask them to fully address evidence in support or in conflict with the idea that the RGG box has an a relevant biologic role to validate their experiments that speak to its function*.*”*

We trust that a genome-wide PAR-CLIP experiment is out of scope of our present manuscript as we demonstrate that the RGG box is involved in G-quartet binding using well-established biochemical assays (Figures 2 and 5) and show that Aven RGG box is required for recruitment of G-quartet containing MLL1, MLL4 and reporter mRNAs to the polysomes and translation as evidenced by simultaneously monitoring expression of corresponding proteins (Figures 4, 7 and 10). Moreover, it appears that during the first round of reviews consensus was reached that a genome-wide PAR-CLIP experiment was indeed out of scope of our study.

*“I realize we previously discussed this and agreed that it would be unreasonable to ask them to do these experiments”*.

Therefore, we were unaware of these issues as they were not brought up in the review and thus we feel that is extremely unfair to bring these issues up in the second round of reviews and use it as a major basis to reject the paper.

In addition, Reviewer#1 states that *“PAR-CLIP experiment but didn't finish it, which could transform their paper (or prove their assertions correct) fairly easily)”*.

Genome-wide PAR-CLIP experiments are cumbersome and require extensive validation (Riley & Steitz, Mol Cell, 49:601-604, 2013). For this reason, we feel that a standalone study is merited to carefully carry out genome-wide PAR-CLIP analysis, and validate and functionally characterize resulting hits (including all possible scenarios such as that some mRNAs may be bound, but not translationally regulated by Aven), rather than just to provide a list of potential Aven mRNA targets in our present study. Therefore, based on the data presented in our present study, we are undertaking genome-wide PAR-CLIP experiments from total and polysomal extracts, wherein in parallel we are performing ribosome profiling study to identify not just mRNAs that are AVEN bound, but also those whose polysome-association and translation are regulated by AVEN. As we hope you will appreciate, this approach is not widely used and thus we need to take significant time and be prudent not to produce yet another list of false positives. We thus anticipate that this study will take several years to complete.

II) Moreover, the reviewer was worried about apparent discrepancies between our Melzer et al., paper, which highlighted the role of N-terminal truncated Aven as the major anti-apoptotic form of the protein.

In contrast to Melzer et al., and consistent with our data, full-length Aven was shown to exhibit anti-apoptotic effects (Chau et al., Mol Cell, 2000; Kutuk et al., Eur J. Cancer, 2010; Eibmann/Mezler et al.,Oncogene, 2012; Ouzounova et al., BMC Genomics, 2013). Furthermore, Melzer et al. (Eibmann/Mezler et al., Oncogene, 2012) published another paper where overexpression of full-length Aven driven by the *lck* promoter opposes apoptosis in primary thymocytes. Accordingly, we did not observe the truncated form of Aven in MOLT-4 and CCRF-CEM cells (Figure 6). Taken together, these findings clearly indicate that Aven plays multiple mutually not excluding, and likely cell type specific, roles in promoting cell proliferation and survival. Indeed, we present a hitherto unprecedented role of Aven in promoting proliferation and long-term survival of leukemic cells (Aven is overexpressed in the vast majority of leukemia) by bolstering the translation of MLL1 and MLL4 which as we demonstrate using multiple experimental approaches is dependent on the integrity of the RGG motif (Figures 3, 7 and 10).

Moreover, we performed clonogenic assay that shows that RGG motif is also required for clonogenic survival of U2OS cells (Figure 12). We also modified the Discussion to highlight the discrepancies between Meltzer et al. and our study.

Collectively, we feel that we provide a plethora of evidence for the requirement of the RGG box for AVENs biological function and provide an underpinning mechanism. We also feel that Reviewer #1’s concerns need a significant period of time to complete and merit being addressed in a separate study. For the reasons stated above, we feel that the decision to reject our manuscript should be reconsidered.

Author response image 1.The Aven RGG/RG motif is required for the cell survival. (A)U2OS cells stably expressing either FLAG-Aven, FLAG-AvenΔRGG or siAven were seeded in 10-cm dishes. 24 h following plating, the cells were treated with the indicated doses of UV in duplicates. The visible colonies were counted at 14 days after treatment, respectively, and normalized to 100% without the treatment. Error bars represent SEM (n = 4). Significance was measured by ANOVA followed by post hoc comparison using Tukey test. **p*< 0.05, ***p*< 0.01. **(B)** Western blots to confirm the expression of FLAG-Aven, FLAG-AvenΔRGG and the knock down of endogenous Aven. Tubulin was used as a loading control.**DOI:**
http://dx.doi.org/10.7554/eLife.06234.022

[Editors’ note: the author responses to the re-review follow.]

Reviewer #1 (abridged):

*[…] As I wrote previously, I think the Richard group has “skipped the step, or are avoiding the issue, of establishing that the RGG box is important […]. They do have good data for a lot of this now, except for the importance of the RGG box in the functions of Aven that make it interesting*.*”*

The following text in the Discussion addresses the RGG box and the Melzer paper: “It was reported that an N-terminal […] including those encoding MLL1 and MLL4 required for cell survival.”

Reviewer #3:

*1) It seems disingenuous to call the photocrosslinking IP experiments presented in this paper “PAR-CLIP”. The use of photoactivatable nucleotides to crosslink proteins to RNA to assist in recovery by IP preceded the paper by Hafner et al. by many years. PAR-CLIP, as published, is touted as a transcriptome-wide approach*.

We have replaced PAR-CLP with photocrosslinking IP experiments. We also cite the original paper Cahill et al., 2000 (Steitz), rather than the Hafner paper.

*2) The description of the RGG box as an “RNA Recognition Motif” (in the Introduction) could be misleading, as the RRM is a specific RNA-binding domain that is not the same as the RGG box. Though many proteins contain both*.

We removed “RNA Recognition Motif” and edited the sentence to read: “The RGG/RG motif, also called RGG box, was shown to bind RNA…”.

*3) The rationale behind the selection of SMN and TDRD3 for further study was not clear to this reviewer. Have either been implicated in osteosarcoma or myeloid leukemia? There are 71 Tudor domain proteins in humans according to EMBL SMART database*.

We added the following to the Results section: “Although there are many Tudor domain containing proteins, methylated RGG/RG motifs are known to interact specifically with the Tudor domains of TDRD3, SMN and SPF30 (77; 31).”

4) Can sc1 RNA bind to Aven if it hasn't been induced to fold into a G-quartet?

This was not tested.

5) What fraction of the proteins associated with Aven in the SILAC experiments are RNA-dependent?

We did not perform a RNase control for this experiment. However, we added the following statement in the Results section: “Of the 146 proteins enriched with Aven, but not AvenΔRGG, ∼23% were ribosomal proteins and ∼10% were RNA binding proteins, their association is likely RNA-dependent (Figure 4—figure supplement 1).”

*6) I am surprised that real-time PCR was used to detect enrichment of the G-quadruplex structures in the photocrosslinked IPs. I would have guessed that the G quadruplex would have prevented efficient reverse transcription. The authors should comment on this point*.

The fold induction observed between Aven and the G4 sequences are likely under represented due to the fact that the crosslinks impede the reverse transcription reactions (45)

*7) The significant digits in*
Figure 2
*should be re-evaluated*.

The significant digits were verified, as suggested.